# RoSTE: An Efficient Quantization-Aware Supervised Fine-Tuning Approach for Large Language Models

**Quan Wei** [* 1]  **Chung-Yiu Yau** [* 2]  **Hoi-To Wai** [2]  **Yang (Katie) Zhao** [1]  **Dongyeop Kang** [3]  **Youngsuk Park** [4]  **Mingyi Hong** [1]

## Abstract

Supervised fine-tuning is a standard method for adapting pre-trained large language models (LLMs) to downstream tasks. Quantization has been recently studied as a post-training technique for efficient LLM deployment. To obtain quantized fine-tuned LLMs, conventional pipelines would first fine-tune the pre-trained models, followed by post-training quantization. This often yields suboptimal performance as it fails to leverage the synergy between fine-tuning and quantization. To effectively realize low-bit quantization of weights, activations and KV caches in LLMs, we propose an algorithm named Rotated Straight-Through-Estimator (RoSTE), which combines quantization-aware supervised fine-tuning (QA-SFT) with an adaptive rotation strategy that identifies an effective rotation configuration to reduce activation outliers. We provide theoretical insights on RoSTE by analyzing its prediction error when applied to an overparameterized least square quantized training problem. Our findings reveal that the prediction error is directly proportional to the quantization error of the converged weights, which can be effectively managed through an optimized rotation configuration. Experiments on Pythia, Qwen and Llama models of different sizes demonstrate the effectiveness of RoSTE. Compared to existing post-SFT quantization baselines, our method consistently achieves superior performances across various tasks and different LLM architectures. Our code is available at https://github.com/OptimAI-Lab/RoSTE.

[*]Equal contribution  [1]Department of Electrical and Computer Engineering, University of Minnesota, USA. [2]Department of Systems Engineering and Engineering Management, The Chinese University of Hong Kong, Hong Kong SAR of China. [3]Department of Computer Science and Engineering, University of Minnesota, USA. [4]Amazon Web Services, USA. Correspondence to: Yang (Katie) Zhao <yangzhao@umn.edu>.

*Proceedings of the 42nd International Conference on Machine Learning*, Vancouver, Canada. PMLR 267, 2025. Copyright 2025 by the author(s).

## 1. Introduction

LLMs have shifted a significant step toward achieving artificial general intelligence (Bubeck et al., 2023) and exhibit remarkable capabilities across different domains, including text generation (Anil et al., 2023; Touvron et al., 2023; Thoppilan et al., 2022), code generation (Chen et al., 2021; Austin et al., 2021; Li et al., 2022), and mathematical problem-solving (Cobbe et al., 2021; Trinh et al., 2024; Wei et al., 2022; Lewkowycz et al., 2022). To adapt LLMs to various applications and scenarios, supervised fine-tuning (SFT) is a standard approach, enabling models to leverage diverse training data and align with specific tasks based on pre-trained models.

While fine-tuned models excel in domain-specific tasks, their substantial computational and storage demands present challenges for efficient deployment, particularly in resource-constrained environments (Xu et al., 2024a). To address these limitations, various model compression techniques have been developed, including quantization (Lin et al., 2023; Frantar et al., 2022), pruning (Ma et al., 2023; Sun et al., 2023), distillation (Xu et al., 2024b), and low-rank approximation (Wang et al., 2024a; Yuan et al., 2023). Among these, quantization is particularly effective for compressing LLMs, as it significantly reduces memory consumption, inference latency and power usage. Additionally, its compatibility with specialized hardware accelerators enhances its practical deployment across diverse devices. Quantization techniques generally fall into two categories: post-training quantization (PTQ) and quantization-aware training (QAT). PTQ is well-suited for quick deployment with minimal resources but often sacrifices accuracy in low-bit settings. In contrast, QAT achieves effective compression with minimal performance loss but requires retraining the entire LLM on a large corpus, incurring substantial computational costs.

For efficient deployment of task-specific LLMs, combining quantization with fine-tuning techniques offers a promising solution. A straightforward approach to obtaining quantized fine-tuned LLMs involves a two-step process: first fine-tune the pre-trained models, then apply quantization. However, applying quantization through PTQ in the second step often degrades the performance of the fine-tuned models, while

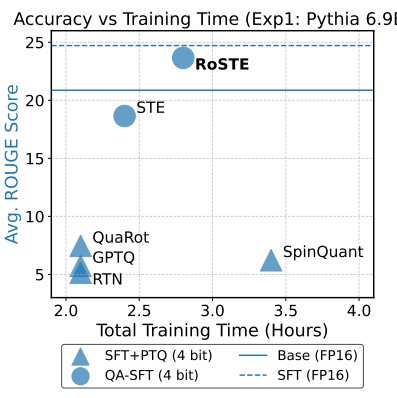 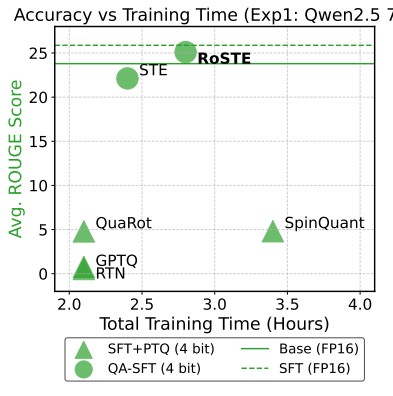 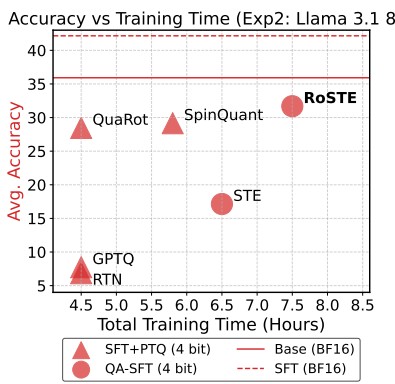

*Figure 1.* RoSTE surpasses the performance of SOTA quantization methods on fine-tuning benchmark. Horizontal axis represents the total amount of hours needed to fine-tune pre-trained LLMs on a server of $8 \times$ A100 NVIDIA GPUs.

QAT introduces an additional training phase, substantially increasing computational costs. Treating fine-tuning and quantization as separate steps can lead to suboptimal results, as it fails to exploit the synergy between these processes.

This work presents one of the first studies on *quantization-aware supervised fine-tuning (QA-SFT)* to obtain effective fine-tuned and quantized LLM through a single training phase. To maximize the hardware capability of modern GPUs, we concentrate on designs utilizing the 4-bit quantization of weights, activations, and KV cache in LLMs. However, low-bit quantization presents a major challenge on models with weight and activation outliers: they expand the quantization range and increase the quantization error, degrading the quantized model accuracy. This partly explains why high-performance data-free QAT methods (Liu et al., 2023) fail at 4-bit activation quantization.

The first key aspect of our work is to leverage rotation-based quantization in QA-SFT. Our work is inspired by recent findings on rotation-based PTQ methods (Ashkboos et al., 2024b; Liu et al., 2024), which demonstrate that applying offline and online rotations to linear projection layers and KV caches in LLMs effectively mitigates weight and activation outliers in post-trained models. However, it is not clear whether one-shot PTQ approaches should be performed before or after fine-tuning. To address this, we propose **a joint training method combining an adaptive selection of rotation matrices and QA-SFT**. The second key aspect of our work is to utilize a bilevel optimization formulation that **simultaneously tackles QA-SFT and selects the rotation matrices based on the weights and activations**.

This paper proposes the Rotated Straight-Through-Estimator (RoSTE) algorithm that integrates the aforementioned ingredients. Our contributions are summarized as:

- We introduce a novel SFT training problem that directly optimizes quantized weights and rotation matrices within a single model architecture. To tame the non-smooth

manifold optimization, we propose a bilevel optimization formulation, where *upper level subproblem* optimizes weight matrices, while *lower level subproblem* employs a surrogate loss to guide the selection of rotation matrix.

- To tackle the bilevel QA-SFT optimization, we propose the RoSTE algorithm which alternates between (i) a QAT subroutine incorporating a rotation-enabled straight-through-estimator (STE) update, and (ii) a low complexity heuristic for selecting rotation matrices based on the random Walsh-Hadamard matrix.

- We provide a theoretical analysis of the benefits of rotation-enabled quantization in QA-SFT by examining the prediction error resulted from the QAT stage of RoSTE. This analysis directly motivates the use of quantization error-based surrogate loss and justifies the adoption of the low-complexity Walsh-Hadamard rotation.

We conduct experiments on fine-tuning Pythia, Qwen and Llama models, demonstrating the effectiveness of RoSTE. An accuracy-vs-training-time plot in Figure 1 illustrates that RoSTE finds quantized fine-tuned models with improved downstream tasks accuracy at the cost of a marginally longer training time. Overall, we believe this is the first work that develops an efficient quantization algorithm for the SFT process, with theoretical justification and practical effectiveness.

### 1.1. Related Works

The quest on quantizing modern LLMs for efficient deployment started with weight-only quantization. GPTQ (Frantar et al., 2022) stood out as a robust postprocessing algorithm that matches the layer output between a quantized model and a full-precision target model. Soon after, a new trend turned to tackling outlier values in weight matrices and activations due to their incompatibility with quantization (Lin et al., 2023; Lee et al., 2023; Chee et al., 2024; Tseng et al., 2024), pushing the limit of accurately quantized models

below 2-bits. While the memory consumption of storing the model parameters is reduced by weight-only quantization, their activations remain in full precision during inference which prohibits the application of long context LLMs on consumer-grade accelerators with limited memory storage.

This motivates the development of weight-activation quantization. It allows weights and activations to be directly multiplied using discrete arithmetic units that accelerate inference and reduce the inference memory requirement. Existing methods can be categorized as follows: (1) mixed-precision quantization (Dettmers et al., 2022; Zhao et al., 2024) that assigns extra bit-widths to outlier values; (2) scaling-based quantization (Xiao et al., 2022; Shao et al., 2023) that employs scaling to balance the representation range between activations and weights; (3) rotated quantization (Ashkboos et al., 2024b; Liu et al., 2024) that utilizes orthogonal transformation to remove activation outliers; (4) knowledge distillation (Liu et al., 2023; Du et al., 2024; Xu et al., 2024c) that re-trains a quantized model to match the behavior of a full-precision target model. Among these methods, rotated quantization methods demonstrate superior performance in 4-bit weight-activation quantization.

## 2. Preliminary

This section provides an overview of two major approaches for achieving efficient quantized LLMs, which are two key ingredients to the proposed RoSTE algorithm.

### 2.1. Post-Training Quantization (PTQ)

The main objective of post-training quantization is to find a quantized model that preserves the behavior of the original model. While sophisticated quantizer designs such as vector quantization (Tseng et al., 2024; Egiazarian et al., 2024) can maintain a rich representation of weight values using $\leq 2$ bits on average, most existing works are limited to weight-only quantization. In contrast, for computationally efficient designs with quantized weights and activation, we focus on uniform quantization that compresses a full-precision tensor into one floating point scaling factor and a set of bit-width limited integers. This scheme is known for its practical efficiency across different modern hardware (Jacob et al., 2018; Ashkboos et al., 2024b).

Formally, the $b$-bits *uniform quantizer* can be expressed as

$$Q(\mathbf{X}) = \left( \underbrace{\mathrm{clamp}_b\left( \left\lfloor \frac{\mathbf{X}}{s(\mathbf{X})} \right\rceil \oplus z(\mathbf{X}) \right)}_{b\text{-bits integer tensor}} \ominus z(\mathbf{X}) \right) s(\mathbf{X})$$

(1)

where $\mathbf{X}$ is a high-precision floating-point tensor; $\lfloor \cdot \rceil$ denotes an element-wise nearest rounding; $\mathrm{clamp}_b(\cdot)$ projects the values to the range of $b$-bits representable integers; $\oplus, \ominus$

represent element-wise addition/subtraction between tensor and scalar. The choice of scaling $s(\mathbf{X}) \in \mathbb{R}$ and shifting $z(\mathbf{X}) \in \mathbb{Z}$ determines the range of which the $b$-bits integer tensor represents. For *symmetric* quantization, we adopt

$$s(\mathbf{X}) = \frac{\max(|\mathbf{X}_{:}|)}{2^{b-1} - 1} c, \quad z(\mathbf{X}) = 0,$$

(2)

$$\mathrm{clamp}_b(\mathbf{X}) = \max\{-2^{b-1}, \min\{\mathbf{X}, 2^{b-1} - 1\}\}$$

(3)

with $c \in (0, 1]$ a constant *clipping factor* used to scale down the representation range so as to mitigate the impact of outlier values. To take advantage of the representation range in tensor with value distribution skewed away from 0, we can adopt *asymmetric* quantization by choosing

$$s(\mathbf{X}) = \frac{\max(\mathbf{X}_{:}) - \min(\mathbf{X}_{:})}{2^b - 1} c, \ z(\mathbf{X}) = \left\lfloor \frac{-\min(\mathbf{X})}{s(\mathbf{X})} \right\rceil,$$

$$\mathrm{clamp}_b(\mathbf{X}) = \max\{0, \min\{\mathbf{X}, 2^b - 1\}\}$$

(4)

The above uniform quantization scheme reduces the memory consumption from storing a $d$-elements 32-bit floating-point tensor $\mathbf{X}$ using $32d$ bits, to storing an integer tensor with its shifting and scaling scalars using $bd + b + 32$ bits.

In practice, we partition a tensor into quantization groups such that each group has its own scaling and shifting $s(\mathbf{X}), z(\mathbf{X})$. Further description of quantizer hyperparameters used in our work will be provided in Appendix D.

**Incoherence Processing via Rotation.** The precision of uniform quantization degrades as the representation range increases, especially when there are outlier values in the full-precision tensor. To reduce the effects of outlier values, incoherence processing was proposed in (Chee et al., 2024) which pre-multiplies an orthogonal matrix to the full-precision tensor prior to quantization, and post-multiplies the transposed orthogonal matrix to recover the original tensor. Later in (Ashkboos et al., 2024b; Tseng et al., 2024), incoherence processing by Walsh-Hadamard rotation is shown to be effective in both uniform quantization and vector quantization for 4-bits weight-activation quantization in LLMs.

We illustrate the idea of incoherence processing by constructing a multi-layer feedforward neural network with activation quantizer $Q_x$ and weight quantizer $Q_w$. The output of the $i$-th linear layer is given by

$$\mathtt{LIN}_i(\mathbf{X}; \mathbf{W}_i^{\star}, \mathbf{R}_i) = \sigma(Q_x(\mathbf{X}\mathbf{R}_i)Q_w(\mathbf{R}_i^{\top}\mathbf{W}_i^{\star}))$$

(5)

where $\mathbf{X}$ is the input activation; $\mathbf{W}_i^{\star}$ is the pre-trained weight matrix; $\mathbf{R}_i$ denotes rotation matrix and $\sigma$ is any activation function. Notice that as $\mathbf{R}_i\mathbf{R}_i^{\top} = \mathbf{I}$, the architecture in (5) is *invariant* to any choice of rotation matrix $\mathbf{R}_i$ when both quantizers $Q_w, Q_x$ are error-free, i.e., $Q_w(\cdot), Q_x(\cdot)$ are the identity map. In general when $Q(x) \neq x$, it has been observed that these rotation matrices suppressed outliers

within each quantizer and preserved the pre-trained model behavior during inference. On the downside, they impose extra memory and computation overhead since rotation is performed during inference within the activation quantizer. Thankfully, these overheads do not counteract the benefits of incoherence processing due to the fast Hadamard CUDA kernels (Ashkboos et al., 2024b; Tseng et al., 2024).

## 2.2. Quantization-Aware Training (QAT)

The main objective of quantization-aware training (QAT) is to directly optimize a quantized neural network using gradient-based methods. From an optimization perspective, this is challenging as the quantization operator is not differentiable. To this regard, straight-through estimator (STE) (Courbariaux et al., 2015; Bai et al., 2018) is a commonly adopted remedy which approximates the Jacobian of quantizer by the identity matrix. During the backward calculation, the derivative of quantizer $Q$ in the chain rule is replaced by

$$\frac{\partial Q(g(\mathbf{X}))}{\partial \mathbf{X}} \approx \frac{\partial g(\mathbf{X})}{\partial \mathbf{X}}, \qquad (6)$$

for any differentiable function $g$. This approximation utilizes the insight that a quantizer behaves like an identity function in low resolution, while tolerating a gradient bias since quantization error persists in high resolution. In practice, STE is known to work well in training quantized neural network models (Li et al., 2017; Yin et al., 2019) as well as LLMs (Liu et al., 2023; Panferov et al., 2025).

These QAT techniques are useful for the sceneario when we consider obtaining a quantized LLM that minimizes the *fine-tuning* objective, as introduced below.

## 2.3. Supervised Fine-Tuning (SFT)

Foundation models that were pre-trained on large unstructured text corpus require fine-tuning to adapt their output behavior for specialized applications such as coding assistants (Chen et al., 2021) and instruction-following conversational chatbot (Ouyang et al., 2022). Towards this goal, Supervised fine-tuning (SFT) resumes the training of a given (pre-trained) model with the data distribution replaced by an application-specific curated dataset (Chung et al., 2024).

In specific, let $\mathcal{D} := \{(\mathbf{x}_i, \mathbf{y}_i)\}_{i=1}^N$ denote the SFT dataset with $N$ samples. For each $i \in [N]$, $\mathbf{x}_i \in \mathcal{X}$ is a sequence of input prompt and $\mathbf{y}_i = (y_{i,0}, \ldots, y_{i,T-1})$ with $y_{i,t} \in \mathcal{Y}$ is a sequence of preferred output tokens. To fine-tune the model with $\mathcal{D}$, we consider minimizing the following SFT loss:

$$\mathcal{L}_{\text{SFT}}(\mathbf{m}(\cdot)) := \mathbb{E}_i \left[ -\sum_{t=0}^{T-1} \log \mathrm{P}(y_{i,t}|\mathbf{x}_i, y_{i,<t}; \mathbf{m}(\cdot)) \right] \quad (7)$$

where the expectation is taken w.r.t. $i \in \{0, \ldots, N-1\}$ with a uniform distribution, $y_{i,<t}$ denotes the sequence of

tokens preceding $y_{i,t}$. The likelihood $\mathrm{P}(y_{i,t}|\mathbf{x}_i, y_{i,<t}; \mathbf{m})$ is the probability of the target token $y_{i,t}$ given the input $\mathbf{x}_i$ and prior context $y_{i,<t}$, as predicted by the model $\mathbf{m}$.

Although it is a natural idea to apply QAT on fine-tuning tasks, existing works such as (Dettmers et al., 2023; Xu et al., 2023) (also see (Lee et al., 2024; Bondarenko et al., 2024) for similar ideas applied to training LLMs) only considered quantization aware adaptation utilizing an additional LoRA architecture. Moreover, they focused on direct quantization without incoherence processing whose performance can be sensitive to outliers. This motivates us to consider integrating QAT with incoherence processing for SFT. In particular, we shall reveal the separate roles of weight matrices and rotation matrices from the perspective of optimization in the next section.

## 3. Proposed Algorithm: RoSTE

This section presents the Rotated Straight-Through-Estimator (RoSTE) algorithm through jointly optimizing the rotation matrices and model parameters. To fix the idea, we parameterize the quantized LLM by the weight matrices $\{\mathbf{W}_i\}_{i=0}^{\ell-1}$ and rotation matrices $\{\mathbf{R}_i\}_{i=0}^{\ell-1}$. Consider the abstraction of an LLM with $\ell$ layers/modules as $\mathbf{m}_Q : \overline{\mathcal{X}} \to \mathbb{R}^{|\mathcal{T}|}$, where $\overline{\mathcal{X}}$ is the set of sequences with variable context length, $\mathcal{T}$ is the set of vocabulary, we denote

$$\mathbf{m}_Q(\overline{\mathbf{x}}; \{\mathbf{W}_i, \mathbf{R}_i\}_{i=0}^{\ell-1}) := \mathtt{NN}(\overline{\mathbf{x}}; \{\mathtt{LIN}_i(\cdot; \mathbf{W}_i, \mathbf{R}_i)\}_{i=0}^{\ell-1}),$$

for any $\overline{\mathbf{x}} \in \overline{\mathcal{X}}$, where $\mathtt{LIN}$ was defined in (5), and $\mathtt{NN}$ denotes the neural network architecture such as transformers[1].

With the above parameterization, an ideal strategy is to consider the optimization problem:

$$\min_{\{\mathbf{W}_i, \mathbf{R}_i\}_{i=0}^{\ell-1}} \mathcal{L}_{\text{SFT}}\big(\mathbf{m}_Q(\,\cdot\,; \{\mathbf{W}_i, \mathbf{R}_i\}_{i=0}^{\ell-1})\big) \qquad (8)$$
$$\text{s.t. } \mathbf{R}_i \mathbf{R}_i^\top = \mathbf{I}, \ i = 0, \ldots, \ell-1,$$

which *simultaneously* selects the weight and rotation matrices under quantization and directly optimizes the SFT objective[2] of the quantized-and-rotated model.

However, tackling (8) can be challenging even with approaches such as alternating optimization. This is because, upon fixing the weight matrices, minimizing the objective function w.r.t. the rotation matrices $\{\mathbf{R}_i\}_{i=0}^{\ell-1}$ involves an intractable manifold optimization while the objective function is non-differentiable due to quantization. Meanwhile, when the rotation matrices are fixed, the minimization problem

---

[1]For simplicity, we excluded the self-attention layers from our notation but the same idea applies to all the layers in the transformer architecture, as demonstrated in (Liu et al., 2024). See Appendix D for the implementation details.

[2]Our approach can be applied on other fine-tuning objectives as well, e.g., (Chen et al., 2024; Li et al., 2024).

---

**Algorithm 1** RoSTE Algorithm

---

1: **Input:** Pre-trained model parameters $\{\mathbf{W}_i^{\mathrm{pt}}\}_{i=0}^{\ell-1}$, step size $\eta > 0$.
2: Initialize $\mathcal{W}^0 = \{\mathbf{W}_i^{\mathrm{pt}}\}_{i=0}^{\ell-1}$.
3: **for** $k = 0, \ldots, K-1$ **do**
4:    /* Rotation configuration */
5:    Find an approximate lower level solution

$$\mathcal{R}^k = \underset{\mathbf{R}_i \in \{\mathbf{H}, \mathbf{I}\}}{\mathrm{argmin}} \; \mathcal{E}(\mathcal{W}^{kT}, \{\mathbf{R}_i\}_{i=0}^{\ell-1}), \qquad (9)$$

   where each $\mathbf{R}_i$ is chosen as either the identity matrix $\mathbf{I}$, i.e., no rotation, or a random Walsh-Hadamard matrix $\mathbf{H}$, generated according to (19).
6:    /* QAT Stage via STE */
7:    **for** $t = 0, \ldots, T-1$ **do**
8:       Draw a mini-batch of training samples $\xi^{kT+t} \subseteq \{0, ..., N-1\}$ uniformly at random and update

$$\mathcal{W}^{kT+t+1} = \mathcal{W}^{kT+t} \qquad (10)$$
$$\overset{\mathrm{s.t.e.}}{-} \eta \, \nabla_{\mathcal{W}} \mathcal{L}_{\mathrm{SFT}}(\mathbf{m}_Q(\cdot; \mathcal{W}^{kT+t}, \mathcal{R}^k); \xi^{kT+t})$$

9:    **end for**
10: **end for**
11: **Output:** Quantized fine-tuned $\mathbf{m}_Q(\cdot; \mathcal{W}^{KT}, \mathcal{R}^{K-1})$.

---

w.r.t. the weight matrices $\{\mathbf{W}_i\}_{i=0}^{\ell-1}$ is similar to standard QAT; see (Liu et al., 2023).

The above obstacle motivated us to consider an alternative formulation (albeit somewhat heuristic) that simplifies the search for $\{\mathbf{R}_i\}_{i=0}^{\ell-1}$ adapted to the weight matrices. This formulation explicitly separates the process of (quantized) model training and the rotation matrix optimization, and leverages a simpler objective function over the rotation matrices. More specifically, we consider:

$$\min_{\{\mathbf{W}_i\}_{i=0}^{\ell-1}} \mathcal{L}_{\mathrm{SFT}}(\mathbf{m}_Q(\cdot; \{\mathbf{W}_i, \mathbf{R}_i^\star\}_{i=0}^{\ell-1})) \qquad (11)$$
$$\mathrm{s.t.} \; \{\mathbf{R}_i^\star\}_{i=0}^{\ell-1} \in \mathrm{argmin}_{\{\mathbf{R}_i\}_{i=0}^{\ell-1}} \; \mathcal{E}(\{\mathbf{W}_i, \mathbf{R}_i\}_{i=0}^{\ell-1})$$
$$\mathrm{s.t.} \; \mathbf{R}_i \mathbf{R}_i^\top = \mathbf{I}, \; i = 0, \ldots, \ell-1,$$

which is a bilevel optimization problem where the lower level optimal rotation matrices $\{\mathbf{R}_i^\star\}_{i=0}^{\ell-1}$ minimize the weight-activation quantization error:

$$\mathcal{E}(\{\mathbf{W}_i, \mathbf{R}_i\}_{i=0}^{\ell-1}) = \sum_{i=0}^{\ell-1} \|Q_w(\mathbf{R}_i^\top \mathbf{W}_i) - \mathbf{R}_i^\top \mathbf{W}_i\|^2 \quad (12)$$
$$+ \frac{1}{n} \sum_{i=0}^{\ell-1} \sum_{j=0}^{n-1} \|Q_x(\mathbf{X}_{i,j} \mathbf{R}_i) - \mathbf{X}_{i,j} \mathbf{R}_i\|^2,$$

for $\mathbf{X}_{i,j}$ representing the input activation of layer $i$ on the $j$-th calibration data sample, e.g., by drawing a subset of size $n$ from $\mathcal{D}$ the fine-tuning dataset.

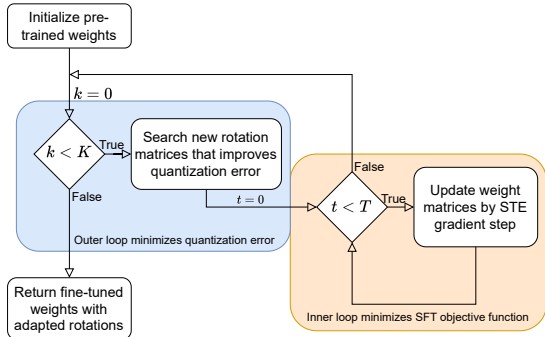

*Figure 2.* The RoSTE algorithm alternates between tackling the lower level problem for rotation configuration and the upper level problem of SFT training using rotation-aware STE.

Motivated to solving the ideal formulation (8), in our reformulation (11), the optimal lower level variable aims at *assisting* the upper level weights so that an STE gradient approximation on $\mathcal{L}_{\mathrm{SFT}}$ w.r.t. $\{\mathbf{W}_i\}_{i=0}^{\ell-1}$ has a smaller bias. However, it remains challenging for us to access the optimal rotation matrices for every iteration of the upper level minimization as solving the lower level problem can still be computationally expensive. In this regard, we propose a *lazy* lower level approximation where the rotation matrices are updated after $T$ iterations of optimizing the weight matrices.

The RoSTE algorithm is now summarized in Algorithm 1 and Fig. 2. The algorithm is akin to alternating optimization and consists of two parts. The first part (cf. line 7–9) pertains to the QAT stage with SFT objective for selecting the weight matrices $\{\mathbf{W}_i\}_{i=0}^{\ell-1}$ under the rotation matrices. Notice that the computation overhead introduced by rotations are insignificant when $\{\mathbf{R}_i\}$ are chosen as the Walsh-Hadamard matrices. The second part (cf. line 5) pertains to the selection of rotation matrices in the lower level optimization which is a non-smooth problem on the manifold. Compared to (8), the lower level objective function (12) can be easily computed through calculating quantized weights and a few mini-batch forward passes on sample data. Furthermore, as we will show in Sec. 4, the random Walsh-Hadamard matrix $\mathbf{H}$ yields an approximate-but-universal solution to minimize $\mathcal{E}(\cdot)$. As such, we propose to approximate the subproblem by limiting the search space to $\mathbf{R}_i \in \{\mathbf{H}, \mathbf{I}\}$ using a random Hadamard matrix $\mathbf{H}$ (Tseng et al., 2024) and perform a (low-complexity) combinatorial search to obtain an approximate lower level solution that decides if the rotation matrix should be applied on each layer. Details of this heuristic implementation can be found in Appendix D.

## 4. Theoretical Insights of RoSTE

This section aims at providing theoretical insights on the RoSTE algorithm that tackles the bilevel problem (11). In particular, we show that the quantization error (12) is a

suitable surrogate loss for optimizing the rotation matrices, provided that the weight matrices are optimized using the STE method as in Algorithm 1. We remark that the SFT objective on quantized LLMs is complicated and possibly untractable for analysis. To concentrate on the insights pertaining to using rotation in the quantized LLMs, we shall introduce a few approximations. We will use $\langle \cdot \mid \cdot \rangle$ to denote inner products of vectors, and $\|\mathbf{x}\|_{\mathbf{K}}^2 = \langle \mathbf{x} \mid \mathbf{K}\mathbf{x} \rangle$ to denote a $\mathbf{K}$-weighted squared norm of vector $\mathbf{x}$ for any square matrix $\mathbf{K}$.

Our setup follows from the literature on analyzing the convergence of SGD for neural networks under the interpolation regime (Ma et al., 2018; Vaswani et al., 2019). To describe it, let us fix the rotation matrices $\{\mathbf{R}_i\}_{i=0}^{\ell-1}$ and consider the QAT stage (cf. line 7–9 in the RoSTE algorithm. Instead of analyzing $\mathcal{L}_{\mathrm{SFT}}(\mathbf{m}_Q(\cdot))$ directly, we consider the quadratic loss function as a simplified objective to draw insights for RoSTE. Moreover, the training dataset consists of samples $(\mathbf{x}_\xi, \mathbf{y}_\xi)$ with a target output token in $\mathbb{R}$ such that $\mathbf{y}_\xi \in \mathcal{Y} \equiv \mathbb{R}$.

For any $\mathbf{m} : \overline{\mathcal{X}} \to \mathbb{R}^{|\mathcal{T}|}$, we now consider the squared prediction error:

$$\widehat{\mathcal{L}}(\mathbf{m}(\cdot)) := \frac{1}{2}\mathbb{E}_\xi \left[ (o(\mathbf{m}(\mathbf{x}_\xi)) - \mathbf{y}_\xi)^2 \right], \qquad (13)$$

in lieu of $\mathcal{L}_{\mathrm{SFT}}(\cdot)$, where $o : \mathbb{R}^{|\mathcal{T}|} \to \mathcal{Y}$ maps the probability distribution over $\mathcal{T}$ to a token.

We further assume that the composite map $o(\mathbf{m}_Q(\cdot))$ is a linear activation-weight quantized model given by

$$o(\mathbf{m}_Q(\mathbf{x}; \mathbf{w}, \mathbf{R})) = \langle Q_x(\mathbf{R}\mathbf{x}) \mid Q_w(\mathbf{R}\mathbf{w}) \rangle, \qquad (14)$$

where $\mathbf{R}$ is a rotation matrix satisfying $\mathbf{R}\mathbf{R}^\top = \mathbf{I}$ and $Q_x, Q_w : \mathbb{R}^d \to \mathbb{R}^d$ are the quantization functions [see Sec. 2.1]. Let $\mathbf{x}_t, \mathbf{y}_t$ be the sample drawn at iteration $t$ in the inner loop update of line 8, Algorithm 1, we have

$$\mathbf{w}^{t+1} = \mathbf{w}^t - \eta \, \boldsymbol{g}_{\mathrm{s.t.e.}}^t \qquad (15)$$
$$\boldsymbol{g}_{\mathrm{s.t.e.}}^t = (\langle Q_x(\mathbf{R}\mathbf{x}_t) \mid Q_w(\mathbf{R}\mathbf{w}^t) \rangle - \mathbf{y}_t)\mathbf{R}^\top Q_x(\mathbf{R}\mathbf{x}_t),$$

where $\eta > 0$ is the step size and we have used the STE approximation $\partial(Q_w(\mathbf{R}\mathbf{w}))/\partial(\mathbf{w}) \approx \mathbf{R}$ when computing the stochastic gradient $\boldsymbol{g}_{\mathrm{s.t.e.}}^t$ at $\mathbf{w}^t$.

Our next endeavor is to study an upper bound on the loss value of quantized model, $\widehat{\mathcal{L}}(\mathbf{m}_Q(\,\cdot\,; \mathbf{w}^T, \mathbf{R}))$, after running the recursion (15) for $T \geq 1$ steps. Define the Gram matrix of the quantized-rotated features by

$$\mathbf{G} := \mathbb{E}\left[ Q_x(\mathbf{R}\mathbf{x}_\xi)Q_x(\mathbf{R}\mathbf{x}_\xi)^\top \right] \qquad (16)$$

and make the following assumptions accordingly:

**Assumption 4.1** (Gram Matrix)**.** There exists constants $\lambda_{\min}, \rho > 0$ such that

$$\mathbf{G}^2 \succeq \lambda_{\min}\mathbf{G}, \quad \sup_{0 \leq t \leq T-1} \|Q_x(\mathbf{R}\mathbf{x}_t)\|_{\mathbf{G}}^2 \leq \rho. \quad (17)$$

The above conditions are mild as $\lambda_{\min}$ is only the smallest non-zero eigenvalue of the Gram matrix $\mathbf{G}$ and $\rho$ exists when the input prompts $\mathbf{x}_t$ are bounded.

**Assumption 4.2** (Interpolation)**.** For any orthogonal matrix $\mathbf{R}$, there exists $\mathbf{w}_{\mathbf{R}}^\star \in \mathbb{R}^d$ such that $\mathbf{y}_\xi = \langle Q_x(\mathbf{R}\mathbf{x}_\xi) \mid \mathbf{w}_{\mathbf{R}}^\star \rangle$ for any $\xi$.

The above assumption requires that the quantized-rotated features $(Q_x(\mathbf{R}\mathbf{x}_\xi), \mathbf{y}_\xi)$ are interpolatable by a full-precision model $\mathbf{w}_{\mathbf{R}}^\star$. This assumption is closely related to the standard interpolation assumption that appeared in the literature on training over-parameterized models (Ma et al., 2018; Vaswani et al., 2019). It is worth noticing that Assumption 4.2 does not require the interpolator $\mathbf{w}_{\mathbf{R}}^\star$ to be in the quantized model parameter space (14).

Define the shorthand notation $\mathbf{m}_{Q,\mathbf{R}}^t := \mathbf{m}_Q(\cdot; \mathbf{w}^t, \mathbf{R})$, we observe the following convergence results for the QAT stage during the RoSTE algorithm:

**Theorem 4.3.** *Under Assumptions 4.1, 4.2 and the step size $\eta = \lambda_{\min}/(6\rho)$, the objective value of the quantized model produced by the recursion (15) is bounded by*

$$\mathbb{E}[\widehat{\mathcal{L}}(\mathbf{m}_{Q,\mathbf{R}}^{t+1})] \leq (1-\mu)^{t+1} \, \widehat{\mathcal{L}}(\mathbf{m}_{Q,\mathbf{R}}^0) \qquad (18)$$
$$+ (6 + 2\mu^{-1}) \sum_{s=0}^{t+1} (1-\mu)^{t-s} \, \mathbb{E}\left[ \|\mathbf{e}(\mathbf{R}\mathbf{w}^s)\|_{\mathbf{G}}^2 \right]$$

*for any $t \geq 0$, where $\mu = \frac{\lambda_{\min}^2}{12\rho}$ and $\mathbf{e}(\mathbf{x}) := Q_w(\mathbf{x}) - \mathbf{x}$.*

See Appendix A for the proof. Our result shows that STE only converges to an inexact solution, which is consistent with previous findings on STE training. For instance, when training models with activation-only quantization, (Yin et al., 2019, Lemma 10) proved that the STE gradient is non-vanishing near local minima. For models with weight-only quantization, (Li et al., 2017, Corollary 1) only showed a convergence guarantee for the full-precision weights but not the quantized weights. In comparison to the prior findings, our result demonstrates the convergence of prediction error with quantized model.

More specifically, suppose the QAT stage of RoSTE is run with $T \gg 1$ inner-loop iterations. Applying the theorem shows that given $\overline{\mathbf{R}}$, the resultant prediction error of model $\mathbf{w}^T$ will be bounded by $\mathcal{O}(\sum_{s=0}^T (1-\mu)^{T-s} \, \mathbb{E}\left[ \|Q_w(\overline{\mathbf{R}}\mathbf{w}^s) - \overline{\mathbf{R}}\mathbf{w}^s\|_{\mathbf{G}}^2 \right])$, i.e., a weighted sum of the weight quantization errors during the QAT process. Due to the exponential weighting $(1-\mu)^{T-s}$, the prediction error is dominated by the weight quantization error of recent iterates. Crucially, the above analysis shows that the rotation matrices play a pivoting role in the performance of QAT. This inspires us to apply $\mathcal{E}(\cdot)$ in (12) to guide us in the selection for optimal rotation matrices, covering the weight quantization error of the rotated weight matrices $\|\mathbf{e}(\mathbf{R}_i^\top \mathbf{W}_i)\|^2$.

**Randomized Rotation Matrices.** Now as we demonstrated that the quantization error is crucial to the prediction performance with the quantized model, we turn our focus to tackling the lower-level subproblem in (11). Notice that minimizing $\mathcal{E}(\cdot)$ w.r.t. the rotation matrix remains challenging. Instead of directly tackling the manifold optimization, our strategy is to apply the random Walsh-Hadamard matrix (Tseng et al., 2024) design as an approximate-yet-universal solution. Consider the random rotation matrix:

$$\mathbf{R}(\zeta) = \mathbf{H}\mathrm{Diag}(\mathbf{r}(\zeta)) \tag{19}$$

where $\mathbf{H} \in \mathbb{R}^{d \times d}$ is a Walsh-Hadamard matrix (Fino & Algazi, 1976) and $\mathbf{r}(\zeta) \in \{-1, 1\}^d$ is a random sign vector. Notice that $\mathbf{R}(\zeta)$ is a binary matrix which favors efficient implementation on GPUs.

We observe the following proposition adapted from (Tseng et al., 2024, Lemma 3.1):

**Proposition 4.4.** *Consider a $b_w$-bits symmetric quantizer $Q_w : \mathbb{R}^d \to \mathbb{R}^d$ [cf. (2), (3) with $c = 1$]. For any $\mathbf{w} \in \mathbb{R}^d$,*

- *with $\mathbf{R} = \mathbf{I}$, it holds that*

$$\|Q_w(\mathbf{w}) - \mathbf{w}\|^2 \le \frac{d \max_i \mathbf{w}_i^2}{4(2^{b_w-1} - 1)^2}. \tag{20}$$

- *with $\mathbf{R} = \mathbf{R}(\zeta)$ from (19), with probability $1 - \delta$ we have*

$$\|Q_w(\mathbf{R}(\zeta)\mathbf{w}) - \mathbf{R}(\zeta)\mathbf{w}\|^2 \le \frac{\log(4d/\delta)}{2(2^{b_w-1} - 1)^2}\|\mathbf{w}\|^2. \tag{21}$$

See Appendix B for the proof.

Observe that the quantization error is $\mathcal{O}(d \max_i \mathbf{w}_i^2)$ without rotation, and is $\mathcal{O}(\|\mathbf{w}\|^2)$ with rotation. Note that the former bound is more sensitive to weight vectors with outliers. In particular, the worst case prediction error in the QAT stage with $\mathbf{R}$ chosen as (19) is strictly better than that for the case with $\mathbf{R} = \mathbf{I}$ (no rotation) if

$$\frac{\log(4d/\delta)}{2}\|\overline{\mathbf{w}}_{\mathbf{R}}\|^2 \le \frac{\max_i(\overline{\mathbf{w}}_{\mathbf{I}i})^2 d}{4}, \tag{22}$$

where $\overline{\mathbf{w}}_{\mathbf{R}}$, $\overline{\mathbf{w}}_{\mathbf{I}}$ are the respective converged solutions of (15). It demonstrates that applying the random rotation matrix in (19) suffices to reduce the quantization error of weight matrices that contain outlier values. To obtain the best performance, we design the RoSTE algorithm such that at the outer loop, it chooses between $\mathbf{H}$ or $\mathbf{I}$ (i.e., no rotation) according to the current weight matrices.

*Remark* 4.5. The analysis in Theorem 4.3 and (22) enables a novel interpretation of the bit-widths in $Q_x$ and $Q_w$ during STE training. On one hand, it is beneficial to increase the bit-width of activation quantization $Q_x$ until Assumption 4.2 is satisfied, and further increasing its bit-width would not improve the prediction performance as the bound (18) only depends on *weight quantization error*. On the other hand, increasing the bit-width of weight quantization always reduces the prediction error as seen in (18), (21). It is also interesting to see that despite adopting low-bit activation quantizers, increasing the dimension $d$ may still allow us to satisfy the interpolation condition Assumption 4.2, under the intuition that kernelized high dimensional features are more likely to be separable (Liang & Rakhlin, 2020). In other words, a neural network with high-dimensional hidden representations can tolerate low-bit quantized activations because the information about $\mathbf{x}_\xi$ retains in the high-dimensional discrete vector $Q_x(\mathbf{R}\mathbf{x}_\xi)$.

## 5. Experiments

We evaluate the performance of the proposed RoSTE algorithm for QA-SFT on two standard sets of open-source models and datasets. For the first experiment (Exp. 1), we fine-tune the pre-trained Pythia 1B/6.9B models (Biderman et al., 2023) and Qwen2.5 0.5B/7B models (Yang et al., 2024) on the Reddit TL;DR Summarization dataset (Huang et al., 2024) with evaluation on the TL;DR test dataset using the ROUGE metric (Lin, 2004). For the second experiment (Exp. 2), we fine-tune the pre-trained Llama 3.1 8B model (Dubey et al., 2024) on the Tulu 3 SFT mixture dataset (Lambert et al., 2024) with real-world downstream task evaluations (Gao et al., 2021). These tasks include TruthfulQA (Lin et al., 2021), MMLU-Pro (Wang et al., 2024b), Big-BenchHard (Suzgun et al., 2022), AGIEval (Zhong et al., 2023), GSM8K (Cobbe et al., 2021), and MATH (Hendrycks et al., 2020).

For the RoSTE algorithm, while we relaxed the lower level as a $\ell$-variable binary combinatorial problem (9), solving this sub-problem has a complexity of $\mathcal{O}(2^\ell)$ which is still intractable for models like Llama 3.1 8B with $\ell = 3 \times 32 + 1$. As a remedy, we estimate the solution of (9) by computing only $\mathcal{E}(\mathcal{W}^{kT}, \{\mathbf{I}\}_{i=0}^{\ell-1})$ and $\mathcal{E}(\mathcal{W}^{kT}, \{\mathbf{H}\}_{i=0}^{\ell-1})$, then determine each layer's $\mathbf{R}_i$ by comparing the quantization error layer-wise. Lastly, we set $K = 1$ where a one-shot rotation configuration adaptation by pre-trained model is found to perform well. We anticipate the performance to further improve with larger $K$ on larger datasets. More implementation details can be found in Appendices C, D.

**Baselines.** Besides the proposed RoSTE algorithm, we compare the performances of LLMs with quantized weight and activation obtained by two streams of baseline approaches. The first stream consists of applying PTQ methods on open-source supervised fine-tuned models in (Huang et al., 2024; Lambert et al., 2024). We reproduce the PTQ benchmarks using round-to-nearest (RTN) quantization, GPTQ (Frantar et al., 2022) , QuaRot (Ashkboos et al., 2024b) and Spin-Quant (Liu et al., 2024). The second set consists of QAT

*Table 1.* **Results on Exp.1.** Accuracies of the 4-bit quantized Pythia 6.9B and Qwen2.5 7B models fine-tuned using the Reddit TL;DR dataset. FP16 and BF16 refer to using 16-bit half-precision floating points and 16-bit brain floating points formats, respectively, and W4A4KV4 refers to using 4-bit quantizations on weights, activation, and KV cache.

| Bit-width | Method | ROUGE-1 | ROUGE-2 | ROUGE-L | ROUGE-LSum | ROUGE (Avg.) |
|---|---|---|---|---|---|---|
| | | Pythia-6.9B | | | | |
| FP16 | Base | 28.81 | 9.45 | 22.29 | 22.91 | 20.87 |
| | SFT | 33.69 | 12.60 | 26.27 | 26.31 | 24.72 |
| W4A4KV4 | RTN | 7.42 | 0.06 | 6.53 | 6.56 | 5.14 |
| | GPTQ | 8.16 | 0.08 | 7.06 | 7.60 | 5.73 |
| | QuaRot | 11.70 | 0.23 | 8.52 | 9.39 | 7.46 |
| | SpinQuant | 8.61 | 0.10 | 8.10 | 8.07 | 6.22 |
| | STE | 28.91 | 9.07 | 22.30 | 22.33 | 20.65 |
| | **RoSTE** | **32.60** | **11.54** | **25.25** | **25.25** | **23.66** |
| | | Qwen2.5-7B | | | | |
| BF16 | Base | 32.72 | 11.82 | 25.18 | 25.42 | 23.79 |
| | SFT | 34.75 | 13.59 | 27.56 | 27.58 | 25.87 |
| W4A4KV4 | RTN | 1.07 | 0.00 | 1.01 | 1.01 | 0.77 |
| | GPTQ | 0.72 | 0.00 | 0.69 | 0.69 | 0.53 |
| | QuaRot | 7.21 | 0.10 | 5.93 | 5.93 | 4.79 |
| | SpinQuant | 6.87 | 0.29 | 5.97 | 6.12 | 4.81 |
| | STE | 30.86 | 10.16 | 23.73 | 23.73 | 22.12 |
| | **RoSTE** | **34.01** | **12.89** | **26.74** | **26.74** | **25.10** |

methods applied on the SFT objective, including STE and RoSTE. The hyperparameters for reproducing our experiment results can be found in the Appendix at Table 4 and 5.

All experiments are conducted on a cluster of 8 NVIDIA A100 GPUs. Details of the training and evaluation settings can be found in Appendix C. Statistics of the training cost (time, memory) can be found in Appendix G.

### 5.1. Accuracy of Quantized Models

For Exp.1, we present the accuracies of 4-bits (weights & activation) quantized, fine-tuned Pythia 6.9B and Qwen2.5 7B in Table 1. On quantizing Pythia 6.9B, the best baseline is STE (without rotation). In comparison, RoSTE produces a quantized model that improves the average ROUGE score by +3.01. It recovers the performance of the full-precision SFT model with a gap of only -1.06 ROUGE score. Similarly on Qwen2.5 7B model, RoSTE improves upon the best baseline by +1.78 ROUGE score, with a gap of -0.77 below the full-precision SFT model. For Exp.2, we present the accuracies of 4-bits (weights & activation) quantized, fine-tuned Llama 3.1 8B in Table 2. Observe that RoSTE improved the average accuracy by +2.56 over the best baseline SpinQuant, despite a gap of -10.47 below the full-precision fine-tuned model.

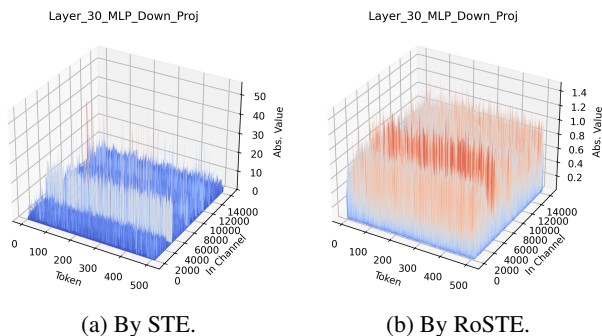

(a) By STE.  (b) By RoSTE.

*Figure 3.* Visualizations of input activations at layer 30 of converged Llama model trained for QA-SFT using STE and RoSTE.

Lastly, in the appendix, we provide additional results of W4A4K4 and W4A8K4 quantization on Pythia 1B in Table 7, Qwen2.5 0.5B in Table 8, and Llama 3.1 8B in Table 9.

### 5.2. Ablation Study

We now concentrate on the effects of optimal rotation configuration as practiced in line 5 of Algorithm 1. In Table 3, we compare the performance on Exp.1 with Pythia 1B when the random Walsh-Hadamard rotation matrix is applied with different strategies. Notice that the adaptive rotation strategy deployed in RoSTE delivered the best performance.

*Table 2.* **Results on `Exp.2`**. Accuracies of the 4-bit quantized Llama 3.1 8B model fine-tuned on the Tulu 3 SFT mixture dataset. `BF16` refers to using 16-bit brain floating points format, and `W4A4KV4` refers to using 4-bit quantizations on weights, activation, and KV cache.

| Bit-width | Method | TruthfulQA | MMLU-Pro | BigBenchHard | AGIEval | GSM8K | Math | Avg. |
|---|---|---|---|---|---|---|---|---|
| BF16 | Base | 28.51 | 19.57 | 62.26 | 30.16 | 56.86 | 18.20 | 35.92 |
| | SFT | 31.82 | 33.07 | 65.67 | 34.86 | 64.89 | 22.66 | 42.16 |
| W4A4KV4 | RTN | 23.01 | 0 | 0 | 17.03 | 1.03 | 0 | 6.85 |
| | GPTQ | 25.34 | 0.02 | 2.55 | 16.48 | 2.05 | 0 | 7.74 |
| | QuaRot | **27.66** | 21.53 | 47.69 | 29.05 | 37.91 | 6.90 | 28.46 |
| | SpinQuant | 26.19 | 21.58 | 49.56 | 28.50 | 38.36 | 10.56 | 29.13 |
| | STE | 26.68 | 9.13 | 24.58 | 17.63 | 22.82 | 1.90 | 17.14 |
| | RoSTE | 26.44 | **25.12** | **52.00** | **30.11** | **44.50** | **11.94** | **31.69** |

*Table 3.* Effects of rotation matrix strategies for STE training in `Exp.1` with Pythia 1B that is W4A4KV4 quantized.

| Rotation Strategy | ROUGE (Avg.) |
|---|---|
| No Rotation | 22.37 |
| Complete Rotation | 13.09 |
| RoSTE (Adaptive Rotation) | **23.07** |

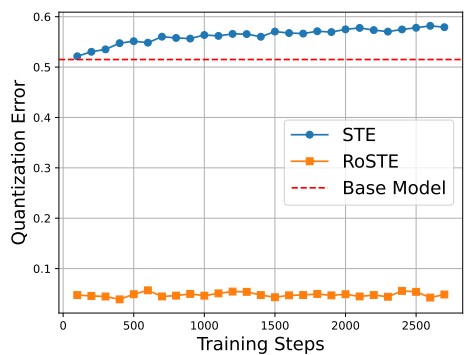

*Figure 4.* The evolution of quantization error (12) against the QAT stage iterations.

When no rotation matrix is applied, we observe a drop in ROUGE score by $-0.70$, and importantly, the complete rotation setting, i.e., applying rotation matrix on every module regardless of whether empirical quantization error is reduced, suffers a drop in ROGUE score by $-9.98$. This shows that while it is beneficial to apply rotation in the STE training, an adaptive strategy such as the one in RoSTE is necessary to guarantee good performance.

Secondly, we take a closer look at the effects of outlier reduction to justify our claims on the use of random Walsh-Hadamard rotation matrices. Fig. 3 compares the distribution of the input activations of fine-tuned models trained by STE and RoSTE at layer 30. We observe that the model produced by RoSTE exhibits no activation outliers, while STE suffers from activation outliers even at convergence.

For a more detailed and comprehensive comparison, see Fig. 6 and 7 in the appendix. Furthermore, Fig. 4 shows the trajectory of the quantization error (12) during training. As expected, we see that the quantization error of RoSTE is much lower than that in STE, thus suggesting a lower bias in the solution obtained [cf. Theorem 4.3].

## 6. Conclusion

This paper proposed the RoSTE algorithm for quantization-aware SFT training with an adaptive rotation strategy. Besides achieving state-of-the-art performance, we also provide theoretical insights to justify the practical efficacy of RoSTE. To the best of our knowledge, this is the first algorithm that leverage adaptive rotation and the fine-tuning objective to produce an accurate quantized model.

## Acknowledgements

M. Hong and Q. Wei are supported in part by NSF grants ECCS-2426064 and CIF-2414372. Q. Wei is also supported by an Amazon Machine Learning Systems Fellowship. C.-Y. Yau and H.-T. Wai are supported by the Shun Hing Institute of Advanced Engineering, CUHK, under Project #MMT-p5-23.

## Impact Statement

This paper presents work whose goal is to advance the field of Machine Learning. There are many potential societal consequences of our work, none of which we feel must be specifically highlighted here.

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

# A. Proof of Theorem 4.3

*Proof.* In this proof, we will use the following equality interchangeably:

$$\widehat{\mathcal{L}}(\mathbf{m}_{Q,\mathbf{R}}^t) = \mathbb{E}\left[\left(\langle Q_x(\mathbf{R}\mathbf{x}_\xi) \mid Q_w(\mathbf{R}\mathbf{w}^t)\rangle - \mathbf{y}_\xi\right)^2\right] = \mathbb{E}\left[\|Q_w(\mathbf{R}\mathbf{w}^t) - \mathbf{w}_{\mathbf{R}}^\star\|_{\mathbf{G}}^2\right] \tag{23}$$

Consider the update rule of our algorithm as $\mathbf{w}^{t+1} = \mathbf{w}^t - \eta(\langle Q_x(\mathbf{R}\mathbf{x}_t) \mid Q_w(\mathbf{R}\mathbf{w}^t)\rangle - \mathbf{y}_t)\mathbf{R}^\top Q_x(\mathbf{R}\mathbf{x}_t)$, we compute

$$\mathbb{E}\left[\left(\langle Q_x(\mathbf{R}\mathbf{x}_\xi) \mid Q_w(\mathbf{R}\mathbf{w}^{t+1})\rangle - \mathbf{y}_\xi\right)^2\right] \tag{24}$$

$$= \mathbb{E}\left[\langle Q_x(\mathbf{R}\mathbf{x}_\xi) \mid Q_w(\mathbf{R}\mathbf{w}^{t+1}) - \mathbf{w}_{\mathbf{R}}^\star + Q_w(\mathbf{R}\mathbf{w}^t) - Q_w(\mathbf{R}\mathbf{w}^t)\rangle^2\right] \tag{25}$$

$$= \mathbb{E}\left[\left(\underbrace{\langle Q_x(\mathbf{R}\mathbf{x}_\xi) \mid Q_w(\mathbf{R}\mathbf{w}^t) - \mathbf{w}_{\mathbf{R}}^\star\rangle}_{p_1^t} + \underbrace{\langle Q_x(\mathbf{R}\mathbf{x}_\xi) \mid Q_w(\mathbf{R}\mathbf{w}^{t+1}) - Q_w(\mathbf{R}\mathbf{w}^t)\rangle}_{p_2^t}\right)^2\right] \tag{26}$$

Now observe that

$$\mathbb{E}\left[(p_2^t)^2\right] \tag{27}$$

$$= \mathbb{E}\left[\|Q_w(\mathbf{R}\mathbf{w}^{t+1}) - Q_w(\mathbf{R}\mathbf{w}^t)\|_{\mathbf{G}}^2\right] \tag{28}$$

$$\leq 3\mathbb{E}\left[\|\mathbf{R}\mathbf{w}^{t+1} - \mathbf{R}\mathbf{w}^t\|_{\mathbf{G}}^2\right] + 3\mathbb{E}\left[\|\mathbf{e}(\mathbf{R}\mathbf{w}^{t+1})\|_{\mathbf{G}}^2\right] + 3\mathbb{E}\left[\|\mathbf{e}(\mathbf{R}\mathbf{w}^t)\|_{\mathbf{G}}^2\right] \tag{29}$$

$$= 3\eta^2\mathbb{E}\left[\|Q_x(\mathbf{R}\mathbf{x}_t)\|_{\mathbf{G}}^2 \langle Q_x(\mathbf{R}\mathbf{x}_t) \mid Q_w(\mathbf{R}\mathbf{w}^t) - \mathbf{w}_{\mathbf{R}}^\star\rangle^2\right] + 3\mathbb{E}\left[\|\mathbf{e}(\mathbf{R}\mathbf{w}^{t+1})\|_{\mathbf{G}}^2\right] + 3\mathbb{E}\left[\|\mathbf{e}(\mathbf{R}\mathbf{w}^t)\|_{\mathbf{G}}^2\right] \tag{30}$$

$$\leq 3\eta^2\rho\mathbb{E}\left[\|Q_w(\mathbf{R}\mathbf{w}^t) - \mathbf{w}_{\mathbf{R}}^\star\|_{\mathbf{G}}^2\right] + 3\mathbb{E}\left[\|\mathbf{e}(\mathbf{R}\mathbf{w}^{t+1})\|_{\mathbf{G}}^2\right] + 3\mathbb{E}\left[\|\mathbf{e}(\mathbf{R}\mathbf{w}^t)\|_{\mathbf{G}}^2\right] \tag{31}$$

where the last inequality is due to the independence $\mathbf{x}_\xi \perp \mathbf{x}_t \perp \mathbf{w}^t$ and uses Assumption 4.1. Furthermore,

$$2\mathbb{E}\left[p_1^t p_2^t\right] \tag{32}$$

$$= 2\mathbb{E}\left[p_1^t \langle Q_x(\mathbf{R}\mathbf{x}_\xi) \mid \mathbf{R}\mathbf{w}^{t+1} - \mathbf{R}\mathbf{w}^t + \mathbf{e}(\mathbf{R}\mathbf{w}^{t+1}) - \mathbf{e}(\mathbf{R}\mathbf{w}^t)\rangle\right] \tag{33}$$

$$= -2\eta\mathbb{E}\left[\langle Q_x(\mathbf{R}\mathbf{x}_\xi) \mid Q_w(\mathbf{R}\mathbf{w}^t) - \mathbf{w}_{\mathbf{R}}^\star\rangle \langle Q_x(\mathbf{R}\mathbf{x}_\xi) \mid Q_x(\mathbf{R}\mathbf{x}_t)\rangle \langle Q_x(\mathbf{R}\mathbf{x}_t) \mid Q_w(\mathbf{R}\mathbf{w}^t) - \mathbf{w}_{\mathbf{R}}^\star\rangle\right]$$

$$\quad + 2\mathbb{E}\left[\langle Q_x(\mathbf{R}\mathbf{x}_\xi) \mid Q_w(\mathbf{R}\mathbf{w}^t) - \mathbf{w}_{\mathbf{R}}^\star\rangle \langle Q_x(\mathbf{R}\mathbf{x}_\xi) \mid \mathbf{e}(\mathbf{R}\mathbf{w}^{t+1}) - \mathbf{e}(\mathbf{R}\mathbf{w}^t)\rangle\right] \tag{34}$$

$$\leq -2\eta\mathbb{E}\left[\|Q_w(\mathbf{R}\mathbf{w}^t) - \mathbf{w}_{\mathbf{R}}^\star\|_{\mathbf{G}^2}^2\right] + \eta\lambda_{\min}\mathbb{E}\left[\|Q_w(\mathbf{R}\mathbf{w}^t) - \mathbf{w}_{\mathbf{R}}^\star\|_{\mathbf{G}}^2\right] \tag{35}$$

$$\quad + \frac{2}{\eta\lambda_{\min}}\mathbb{E}\left[\|\mathbf{e}(\mathbf{R}\mathbf{w}^{t+1})\|_{\mathbf{G}}^2\right] + \frac{2}{\eta\lambda_{\min}}\mathbb{E}\left[\|\mathbf{e}(\mathbf{R}\mathbf{w}^t)\|_{\mathbf{G}}^2\right] \tag{36}$$

$$\leq -\eta\lambda_{\min}\mathbb{E}\left[\|Q_w(\mathbf{R}\mathbf{w}^t) - \mathbf{w}_{\mathbf{R}}^\star\|_{\mathbf{G}}^2\right] + \frac{2}{\eta\lambda_{\min}}\mathbb{E}\left[\|\mathbf{e}(\mathbf{R}\mathbf{w}^{t+1})\|_{\mathbf{G}}^2\right] + \frac{2}{\eta\lambda_{\min}}\mathbb{E}\left[\|\mathbf{e}(\mathbf{R}\mathbf{w}^t)\|_{\mathbf{G}}^2\right] \tag{37}$$

where the last step is due to the independence $\mathbf{x}_\xi \perp \mathbf{x}_t \perp \mathbf{w}^t$ and (17). Therefore, we obtain

$$\mathbb{E}\left[\|Q_w(\mathbf{R}\mathbf{w}^{t+1}) - \mathbf{w}_{\mathbf{R}}^\star\|_{\mathbf{G}}^2\right] \tag{38}$$

$$\leq (1 - \eta\lambda_{\min} + 3\eta^2\rho)\mathbb{E}\left[\|Q_w(\mathbf{R}\mathbf{w}^t) - \mathbf{w}_{\mathbf{R}}^\star\|_{\mathbf{G}}^2\right] + (3 + \frac{2}{\eta\lambda_{\min}})\left(\mathbb{E}\left[\|\mathbf{e}(\mathbf{R}\mathbf{w}^{t+1})\|_{\mathbf{G}}^2\right] + \mathbb{E}\left[\|\mathbf{e}(\mathbf{R}\mathbf{w}^t)\|_{\mathbf{G}}^2\right]\right) \tag{39}$$

$$\overset{(i)}{\leq} (1 - \eta\lambda_{\min}/2)\mathbb{E}\left[\|Q_w(\mathbf{R}\mathbf{w}^t) - \mathbf{w}_{\mathbf{R}}^\star\|_{\mathbf{G}}^2\right] + (3 + \frac{2}{\eta\lambda_{\min}})\left(\mathbb{E}\left[\|\mathbf{e}(\mathbf{R}\mathbf{w}^{t+1})\|_{\mathbf{G}}^2\right] + \mathbb{E}\left[\|\mathbf{e}(\mathbf{R}\mathbf{w}^t)\|_{\mathbf{G}}^2\right]\right) \tag{40}$$

$$\leq (1 - \eta\lambda_{\min}/2)^{t+1}\mathbb{E}\left[\|Q_w(\mathbf{R}\mathbf{w}^0) - \mathbf{w}_{\mathbf{R}}^\star\|_{\mathbf{G}}^2\right] \tag{41}$$

$$\quad + (3 + \frac{2}{\eta\lambda_{\min}})\sum_{k=0}^{t}(1 - \eta\lambda_{\min}/2)^{t-k}\left(\mathbb{E}\left[\|\mathbf{e}(\mathbf{R}\mathbf{w}^{k+1})\|_{\mathbf{G}}^2\right] + \mathbb{E}\left[\|\mathbf{e}(\mathbf{R}\mathbf{w}^k)\|_{\mathbf{G}}^2\right]\right) \tag{42}$$

where $(i)$ uses the step size condition $\eta \leq \lambda_{\min}/(6\rho)$. Choosing the step size $\eta = \lambda_{\min}/(6\rho)$ completes the proof. $\square$

# B. Proof of Proposition 4.4

*Proof.* By the definition of $Q_w$ in (2) and (3), we notice

$$\|Q_w(\mathbf{w}) - \mathbf{w}\|^2 = \left\|s(\mathbf{w})\left\lfloor\frac{\mathbf{w}}{s(\mathbf{w})}\right\rceil - s(\mathbf{w})\frac{\mathbf{w}}{s(\mathbf{w})}\right\|^2 = s(\mathbf{w})^2 \left\|\left\lfloor\frac{\mathbf{w}}{s(\mathbf{w})}\right\rceil - \frac{\mathbf{w}}{s(\mathbf{w})}\right\|^2 \tag{43}$$

$$\overset{(i)}{\leq} s(\mathbf{w})^2 \frac{d}{4} = \frac{d \, \max_i \mathbf{w}_i^2}{4(2^{b_w-1}-1)^2} \tag{44}$$

where $(i)$ is a worst-case error bound of nearest rounding. This proves (20). Further applying (Tseng et al., 2024, Lemma 3.1) in our last step gives (21). $\quad\square$

## C. Details of Experiment Settings

**RoSTE algorithm.** We set the lower level objective function in (12) by drawing $n = 128$ samples from the fine tuning dataset for calibration.

**Hyper-parameters.** We list the training configurations for SFT in (Exp.1) TL;DR summarization and (Exp.2) Tulu 3 experiments as suggested in (Huang et al., 2024; Lambert et al., 2024) in Table 4. For QA-SFT, we sweep through a number of hyper-parameters for STE and RoSTE to obtain the best performance, as listed in Table 5.

*Table 4.* Detailed training settings for SFT in the TL;DR summarization and Tulu 3 experiments.

| Method | | | SFT | | |
|---|---|---|---|---|---|
| Model | Pythia 1B | Pythia 6.9B | Qwen2.5 0.5B | Qwen2.5 7B | Llama 3.1 8B |
| Epoch | 1 | 1 | 1 | 1 | 2 |
| Batch Size (Per GPU) | 16 | 1 | 16 | 1 | 1 |
| Gradient Accumulation | 1 | 16 | 1 | 16 | 16 |
| Optimizer | AdamW | AdamW | AdamW | AdamW | AdamW |
| Learning Rate | 3e-5 | 3e-5 | 5e-5 | 1e-5 | 5e-6 |
| LR Schedule | cosine | cosine | cosine | cosine | linear |
| Warmup Ratio | 0 | 0 | 0 | 0 | 0.03 |
| Max. Seq. Length | 2048 | 2048 | 2048 | 2048 | 1024 |
| # Training Samples | 117k | 117k | 117k | 117k | 100k |

*Table 5.* Detailed training settings and hyper-parameters for QA-SFT in the TL;DR summarization and Tulu 3 experiments.

| Method | | | QA-SFT (i.e., STE or RoSTE) | | |
|---|---|---|---|---|---|
| Model | Pythia 1B | Pythia 6.9B | Qwen2.5 0.5B | Qwen2.5 7B | Llama 3.1 8B |
| Epoch | 1 | 1 | 1 | 1 | 2 |
| Batch Size (Per GPU) | 16 | 1 | 16 | 1 | 1 |
| Gradient Accumulation | 1 | 16 | 1 | 16 | 16 |
| Optimizer | AdamW | AdamW | AdamW | AdamW | AdamW |
| Learning Rate | {3e-5, 6e-6, 3e-6} | {3e-5, 6e-6, 3e-6} | {5e-5, 1e-5, 5e-6} | {5e-5, 1e-5, 5e-6} | {5e-6, 1e-6, 5e-7} |
| LR Schedule | cosine | cosine | cosine | cosine | linear |
| Warmup Ratio | 0 | 0 | 0 | 0 | 0.03 |
| Max. Seq. Length | 2048 | 2048 | 2048 | 2048 | 1024 |
| # Training Samples | 117k | 117k | 117k | 117k | 100k |
| clipping factor | {1, 0.95, 0.9} | {1, 0.95, 0.9} | {1, 0.95, 0.9} | {1, 0.95, 0.9} | {1, 0.95, 0.9} |

**Evalution.** For the TL;DR summarization experiments, all final models are evaluated on the TL;DR test dataset using the ROUGE metric (Lin, 2004), including ROUGE-1, ROUGE-2, ROUGE-L, ROUGE-LSum. For the Tulu 3 experiments, all final models are evaluated on downstream tasks using EleutherAI LM Evaluation Harness (Gao et al., 2021). These tasks include TruthfulQA (Lin et al., 2021), MMLU-Pro (Wang et al., 2024b), BigBenchHard (Suzgun et al., 2022), AGIEval (Zhong et al., 2023), GSM8K (Cobbe et al., 2021), and MATH (Hendrycks et al., 2020). In Table 6, we list the detailed evaluation settings for these downstream tasks as suggested in the Tulu 3 paper (Lambert et al., 2024).

*Table 6.* Details of evaluation settings for the Tulu 3 experiments.

| Benchmark | TruthfulQA | MMLU-Pro | BigBenchHard | AGIEval | GSM8K | Math |
|---|---|---|---|---|---|---|
| # shot | 6 | 0 | 3 | 0 | 8 | 4 |
| Metric | Acc (mc1) | EM | EM | Acc | EM | EM |
| CoT | ✓ | ✗ | ✗ | ✗ | ✓ | ✗ |

## D. Implementation Details of the Rotated-and-Quantized LLM

Our architecture for inserting rotation matrices and quantization on transformer models follows from (Liu et al., 2024). For completeness, an illustration is provided in Figure 5. In the following sections, we describe the details of the RoSTE algorithm under the setting of Exp.1 and Exp.2.

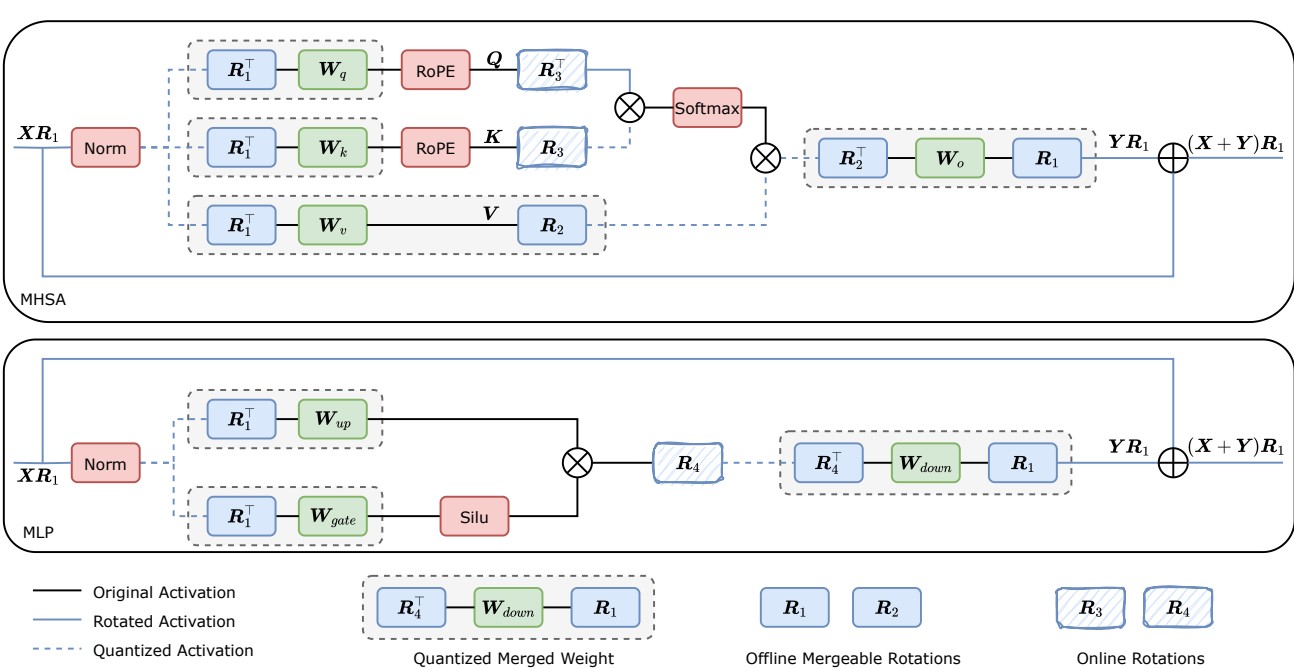

*Figure 5.* An illustration of the rotation workflow in a transformer-based model. $\mathbf{R}_1$ represents the between-block rotation, which eliminates activation outliers between blocks. $\mathbf{R}_2, \mathbf{R}_3, \mathbf{R}_4$ are in-block rotations designed to remove outliers within the MHSA and MLP blocks. Among these, $\mathbf{R}_1, \mathbf{R}_1^\top, \mathbf{R}_2, \mathbf{R}_2^\top, \mathbf{R}_4^\top$ can be merged into weights while $\mathbf{R}_3, \mathbf{R}_3^\top, \mathbf{R}_4$ are not mergeable and serve as online rotations during training and inference.

**Quantization on LLMs.** We adopt the asymmetric uniform quantizer (4) for all the experiments. For instance, to quantize the activations $\mathbf{X}$ of dimensions [*batch size*, *sequence length*, *embedding size*], we employ *per-token* quantization such that each embedding $\mathbf{X}_{ij}$ forms a quantization group. To quantize the linear weight values $\mathbf{W}$, we employ *per-channel* quantization such that each $i$-th output channel's weights $\mathbf{W}_i$ form a quantization group.

**Modifying Normalization Layer.** We modify the model to maintain computational invariance before and after applying rotation. This requires ensuring that there are no mean subtraction, scaling, or shifting operations in the normalization module. For models with LayerNorm, such as Pythia, the process involves absorbing the mean subtraction operation into the weight matrix before LayerNorm and absorbing the LayerNorm scaling and shifting parameters into the weight matrix after the LayerNorm layer (Ashkboos et al., 2024a). Similarly, for models using RMSNorm, such as Llama, this can be achieved by absorbing the RMSNorm scaling parameter into the weight matrix immediately following the RMSNorm layer.

**Between-Block & In-Block Rotation.** We perform between-block rotation $\mathbf{R}_1$ to eliminate the activation outliers between blocks. As illustrated in Fig. 5, $\mathbf{R}_1$ is applied to all linear layers in MHSA and MLP blocks. In particular, the weight

matrices in the Q, K, and V projection layers of MHSA, as well as the Up and Gate projection layers of MLP, are rotated along with their corresponding input activations to preserve computational invariance. Similarly, the weight matrices in the O projection layer of MHSA and the Down projection layer of MLP, along with their corresponding outputs, are also rotated using $\mathbf{R}_1$. Additionally, we also rotate the embedding and lm_head layers so that the final output of the model will be identical to the original model. Next, we perform in-block rotations $\mathbf{R}_2, \mathbf{R}_3, \mathbf{R}_4$ to eliminate the activation outliers within blocks. Specially, $\mathbf{R}_2$ is applied to the Value and the O projection layer of MHSA. $\mathbf{R}_3$ works for the Query and Key. $\mathbf{R}_2$ and $\mathbf{R}_3$ can remove the activation outliers for KV caches. We apply $\mathbf{R}_4$ to the Down projection layer of MLP.

$\mathbf{R}_1, \mathbf{R}_2$ are offline mergeable rotations, which can be merged into the weight matrices before training. $\mathbf{R}_3, \mathbf{R}_4$ are online rotations, which are implemented in the fast Hadamard kernel and can be seen as a layer dynamically rotating the input activation. This online operation is highly efficient by leveraging the fast Hadamard CUDA kernel, resulting in negligible overhead during both training and inference.

## E. Impact of Rotation on Different Models

Fig. 6 showcases the effects of (random Walsh-Hadamard) rotation applied to several exemplary layers in Pythia and Llama models, and demonstrates that sometimes applying the rotation can lead to undesirable results where new outlier values emerge. Fig. 7 presents a comprehensive view of the effects of applying rotations to the weights, activation, and KV cache of different layers. Notice that the RoSTE algorithm only applies rotation when a reduction of quantization error is observed in the respective layers. Moreover, from the figure we observe that in general, the last layers of Pythia model do not benefit from applying rotation, while the rotation effects on Llama model are generally beneficial.

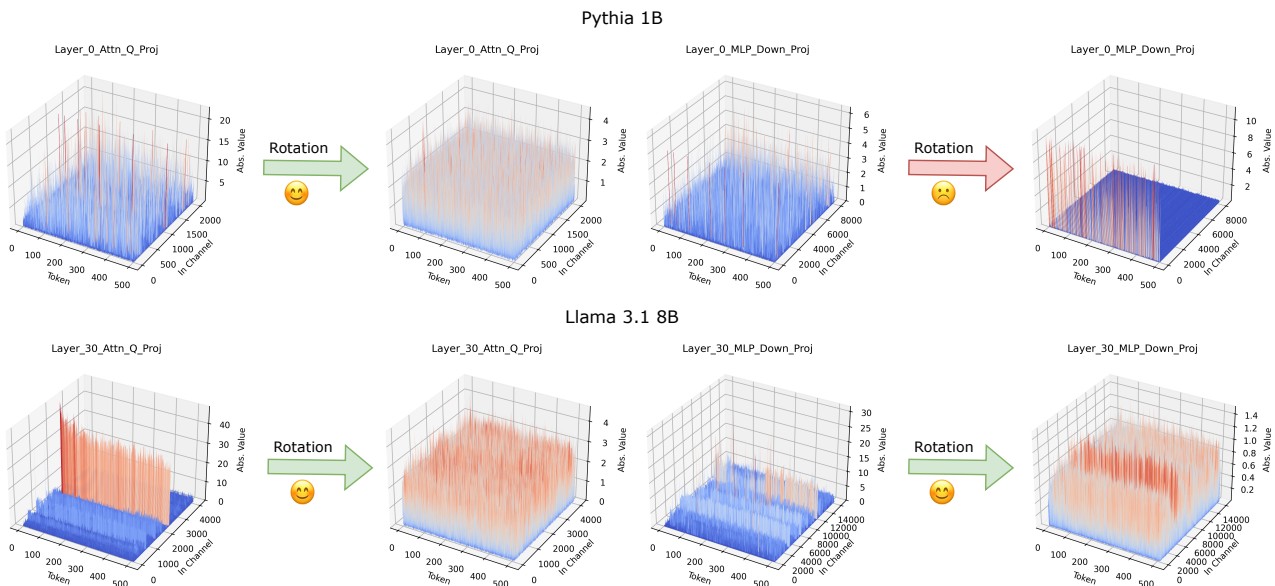

*Figure 6.* Visualizations of Input Activations in Pythia and Llama Models before and after rotation.

We conjecture that several architectural differences between Pythia and Llama contribute to this discrepancy. First, Pythia does not utilize Gated Linear Units (GLU) in its MLP layers, a feature that is integral to Llama. Second, Pythia employs layer normalization (LayerNorm) instead of root mean square normalization (RMSNorm) which is used in Llama. Finally, Pythia adopts a parallel residual connection for attention and feed-forward layers, in contrast to the sequential residual connection found in Llama.

## F. Additional Experiments

We show additional experiment results for the resultant accuracies of fine-tuning the LLMs with different configurations of quantization parameters. Particularly, the results for `Exp.1` on the Pythia models can be found in Table 7, and the results for `Exp.2` on the Pythia models can be found in Table 9. We observe consistent improvements with the RoSTE algorithm.

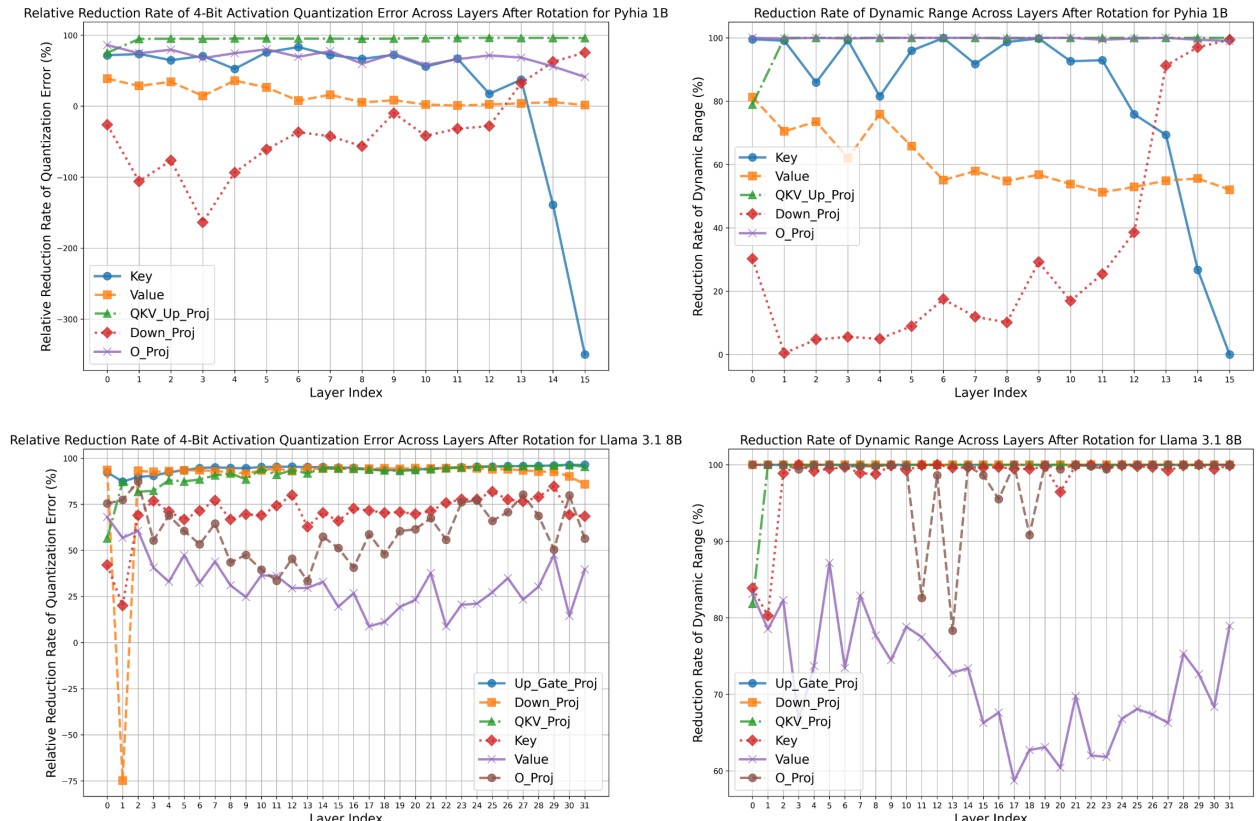

*Figure 7.* Effects of incoherence processing using rotation matrices on *different layers* of Pythia and Llama models using the pre-trained weights. (Left) Relative reduction rates of quantization error, calculated as $\frac{\text{Error w/o rotation} - \text{Error w/ rotation}}{\text{Error w/o rotation}} \times 100\%$. Note that the reduction rate can be negative if the rotation is not beneficial. (Right) Reduction rate of dynamic ranges of the activations after rotation.

Additional comparisons to QLoRA (Dettmers et al., 2023), LLM-QAT (Liu et al., 2023) and DuQuant (Lin et al., 2024) are provided in some setups, for instance, we observed a significant performance degradation on DuQuant when the KV cache is quantized below 8 bits in Table 8.

*Table 7.* Additional experiments for `Exp.1` with different bit-width configurations and different model sizes.

| Bit-width | Method | ROUGE-1 | ROUGE-2 | ROUGE-L | ROUGE-LSum | ROUGE (Avg.) |
|---|---|---|---|---|---|---|
| | | | Pythia-1B | | | |
| FP16 | Base | 22.40 | 5.73 | 17.35 | 17.59 | 15.77 |
| | SFT | 32.80 | 11.84 | 25.49 | 25.50 | 23.91 |
| W4A4KV4 | RTN | 6.05 | 0.06 | 5.21 | 5.67 | 4.25 |
| | GPTQ | 10.16 | 0.30 | 8.41 | 8.84 | 6.93 |
| | LLM-QAT | 19.71 | 4.03 | 15.82 | 15.83 | 13.85 |
| | QuaRot | 16.57 | 1.66 | 13.61 | 13.70 | 11.39 |
| | SpinQuant | 13.52 | 0.40 | 11.21 | 11.10 | 9.06 |
| | QLoRA ($r = 64$) | 22.58 | 5.87 | 17.48 | 17.71 | 15.91 |
| | STE | 31.03 | 10.44 | 24.01 | 24.01 | 22.37 |
| | RoSTE (ours) | **31.80** | **11.03** | **24.71** | **24.71** | **23.07** |
| W4A8KV4 | RTN | 24.19 | 6.94 | 19.29 | 19.13 | 17.39 |
| | GPTQ | 29.77 | 9.81 | 23.38 | 23.50 | 21.52 |
| | LLM-QAT | 29.54 | 9.60 | 23.08 | 23.08 | 21.33 |
| | QuaRot | 30.14 | 9.24 | 22.97 | 23.03 | 21.35 |
| | SpinQuant | 30.37 | 9.73 | 23.15 | 23.43 | 21.67 |
| | STE | 32.44 | 11.48 | 25.24 | 25.24 | 23.60 |
| | RoSTE (ours) | **32.67** | **11.61** | **25.37** | **25.37** | **23.76** |
| | | | Pythia-6.9B | | | |
| FP16 | Base | 28.81 | 9.45 | 22.29 | 22.91 | 20.87 |
| | SFT | 33.69 | 12.60 | 26.27 | 26.31 | 24.72 |
| W4A4KV4 | RTN | 7.42 | 0.06 | 6.53 | 6.56 | 5.14 |
| | GPTQ | 8.16 | 0.08 | 7.06 | 7.60 | 5.73 |
| | LLM-QAT | 18.73 | 3.71 | 15.31 | 15.01 | 13.19 |
| | QuaRot | 11.70 | 0.23 | 8.52 | 9.39 | 7.46 |
| | SpinQuant | 8.61 | 0.10 | 8.10 | 8.07 | 6.22 |
| | QLoRA ($r = 64$) | 27.92 | 8.91 | 21.97 | 22.00 | 20.20 |
| | STE | 28.91 | 9.07 | 22.30 | 22.33 | 20.65 |
| | RoSTE (ours) | **32.60** | **11.54** | **25.25** | **25.25** | **23.66** |
| W4A8KV4 | RTN | 21.77 | 5.31 | 17.31 | 17.22 | 15.40 |
| | GPTQ | 32.42 | 10.71 | 24.56 | 24.59 | 23.07 |
| | LLM-QAT | 29.24 | 9.16 | 22.64 | 22.64 | 20.92 |
| | QuaRot | 26.08 | 8.17 | 20.97 | 20.98 | 19.05 |
| | SpinQuant | 31.69 | 10.70 | 24.69 | 24.68 | 22.94 |
| | STE | 33.05 | 11.94 | 25.58 | 25.61 | 24.05 |
| | RoSTE (ours) | **33.18** | **12.05** | **25.86** | **25.88** | **24.24** |

Table 8. Additional experiments for Exp.1 with different bit-width configurations and different model sizes.

| Bit-width | Method | ROUGE-1 | ROUGE-2 | ROUGE-L | ROUGE-LSum | ROUGE (Avg.) |
|---|---|---|---|---|---|---|
| | | | | Qwen2.5-0.5B | | |
| BF16 | Base | 23.79 | 6.63 | 18.46 | 18.56 | 16.86 |
| | SFT | 32.58 | 11.93 | 25.53 | 25.55 | 23.90 |
| W4A4KV4 | RTN | 10.04 | 0.37 | 8.15 | 8.34 | 6.73 |
| | GPTQ | 12.53 | 0.92 | 10.08 | 10.50 | 8.51 |
| | QuaRot | 9.94 | 0.57 | 8.18 | 8.38 | 6.67 |
| | SpinQuant | 12.16 | 1.22 | 10.69 | 10.72 | 8.70 |
| | DuQuant | 4.05 | 0.09 | 3.53 | 3.58 | 2.81 |
| | QLoRA ($r = 64$) | 24.88 | 7.18 | 19.28 | 19.43 | 17.69 |
| | STE | 29.97 | 9.92 | 23.39 | 23.39 | 21.67 |
| | RoSTE (ours) | **30.75** | **10.44** | **23.96** | **23.96** | **22.28** |
| W4A8KV4 | RTN | 9.51 | 1.06 | 9.02 | 8.90 | 7.12 |
| | GPTQ | 9.53 | 1.04 | 8.80 | 8.73 | 7.03 |
| | QuaRot | 8.24 | 1.25 | 7.51 | 7.23 | 6.06 |
| | SpinQuant | 9.10 | 1.11 | 8.31 | 8.12 | 6.66 |
| | DuQuant | 3.91 | 0.06 | 3.56 | 3.53 | 2.77 |
| | STE | 32.14 | 11.50 | 25.18 | 25.18 | 23.50 |
| | RoSTE (ours) | **32.31** | **11.79** | **25.37** | **25.38** | **23.71** |
| W4A4KV8 | QuaRot | 29.34 | 9.08 | 22.21 | 22.15 | 20.70 |
| | DuQuant | 30.22 | 10.25 | 23.17 | 23.20 | 21.71 |
| | | | | Qwen2.5-7B | | |
| BF16 | Base | 32.72 | 11.82 | 25.18 | 25.42 | 23.79 |
| | SFT | 34.75 | 13.59 | 27.56 | 27.58 | 25.87 |
| W4A4KV4 | RTN | 1.07 | 0.00 | 1.01 | 1.01 | 0.77 |
| | GPTQ | 0.72 | 0.00 | 0.69 | 0.69 | 0.53 |
| | QuaRot | 7.21 | 0.10 | 5.93 | 5.93 | 4.79 |
| | SpinQuant | 6.87 | 0.29 | 5.97 | 6.12 | 4.81 |
| | DuQuant | 0.00 | 0.00 | 0.00 | 0.00 | 0.00 |
| | QLoRA ($r = 64$) | 32.22 | 11.41 | 24.75 | 24.89 | 23.32 |
| | STE | 30.86 | 10.16 | 23.73 | 23.73 | 22.12 |
| | RoSTE (ours) | **34.01** | **12.89** | **26.74** | **26.74** | **25.10** |
| W4A8KV4 | RTN | 5.73 | 0.23 | 4.72 | 4.74 | 3.86 |
| | GPTQ | 7.48 | 0.27 | 6.22 | 6.36 | 5.08 |
| | QuaRot | 5.62 | 0.15 | 5.08 | 5.14 | 3.99 |
| | SpinQuant | 0.64 | 0.30 | 5.64 | 5.81 | 4.54 |
| | DuQuant | 0.24 | 0.00 | 0.24 | 0.24 | 0.18 |
| | STE | 34.44 | 13.29 | 27.16 | 27.17 | 25.52 |
| | RoSTE (ours) | **34.58** | **13.46** | **27.34** | **27.35** | **25.68** |
| W4A4KV8 | QuaRot | 31.96 | 10.98 | 24.73 | 24.88 | 23.13 |
| | DuQuant | 33.47 | 12.13 | 25.28 | 25.30 | 24.05 |

*Table 9.* Additional experiments for Exp.2 on different bit-width configurations.

| Bit-width | Method | TruthfulQA | MMLU-Pro | BigBenchHard | AGIEval | GSM8K | Math | Avg. |
|---|---|---|---|---|---|---|---|---|
| FP16 | Base | 28.51 | 19.57 | 62.26 | 30.16 | 56.86 | 18.20 | 35.93 |
| | SFT | 31.82 | 33.07 | 65.67 | 34.86 | 64.89 | 22.66 | 42.16 |
| W4A4KV4 | RTN | 23.01 | 0 | 0 | 17.03 | 1.03 | 0 | 6.85 |
| | GPTQ | 25.34 | 0.02 | 2.55 | 16.48 | 2.05 | 0 | 7.74 |
| | QuaRot | **27.66** | 21.53 | 47.69 | 29.05 | 37.91 | 6.90 | 28.46 |
| | SpinQuant | 26.19 | 21.58 | 49.56 | 28.50 | 38.36 | 10.56 | 29.13 |
| | STE | 26.68 | 9.13 | 24.58 | 17.63 | 22.82 | 1.90 | 17.14 |
| | RoSTE (ours) | 26.44 | **25.12** | **52.00** | **30.11** | **44.50** | **11.94** | **31.69** |
| W4A8KV4 | RTN | 28.76 | 19.29 | 42.96 | 27.75 | 28.66 | 7.84 | 25.88 |
| | GPTQ | 28.52 | 25.54 | 46.38 | 29.26 | 48.60 | 0.02 | 29.72 |
| | QuaRot | 27.42 | 26.78 | 53.79 | 32.01 | 49.20 | 12.72 | 33.65 |
| | SpinQuant | 28.15 | 26.66 | 55.74 | 32.01 | 52.16 | 15.38 | 35.02 |
| | STE | 29.62 | 24.09 | 54.62 | 29.44 | 52.62 | 4.08 | 32.41 |
| | RoSTE (ours) | **30.84** | **28.23** | **59.25** | **34.03** | **56.94** | **16.88** | **37.70** |

# G. Statistics of Training Cost

Table 10 presents additional statistics for the training costs when using RoSTE and other benchmark algorithms. We observe that while achieving better performance, RoSTE requires only similar amount of computation costs compared to benchmarked algorithms.

*Table 10.* Training time and peak GPU memory consumption for obtaining a quantized fine-tuned Qwen2.5 7B from its pre-trained checkpoint on a server of $8 \times$ A100.

| Bit-width | Method | Training Time (hours) | Peak Memory (GB) |
|---|---|---|---|
| FP16 | SFT | 2.1 | 300 |
| | LoRA ($r = 64$) | 0.55 | 173 |
| W4A4KV4 | SFT $\rightarrow$ GPTQ | 2.1 $\rightarrow$ 0 | 300 $\rightarrow$ 0 |
| | SFT $\rightarrow$ QuaRot | 2.1 $\rightarrow$ 0 | 300 $\rightarrow$ 0 |
| | SFT $\rightarrow$ SpinQuant | 2.1 $\rightarrow$ 1.3 | 300 $\rightarrow$ 263 |
| | QLoRA ($r = 64$) | 0.83 | 98 |
| | STE | 2.4 | 317 |
| | RoSTE | 2.8 | 318 |

