# OpenReview forum: "RoSTE: An Efficient Quantization-Aware Supervised Fine-Tuning Approach for Large Language Models"
_ICML.cc/2025/Conference — ICML 2025 poster_

### Official Review · Reviewer_mQhc · 2025-03-12

**Overall Recommendation:** 4

**Summary:**

This paper introduces RoSTE, a method that combines rotation-based transformation with Quantization-Aware Training to improve the efficiency of the SFT process.

## update after rebuttal

Thanks for providing the additional results. I will raise my score accordingly. Please be sure to include these experimental results and analysis during the rebuttal period in your next version.

**Claims And Evidence:**

The claims are well-supported by clear and convincing evidence.

**Essential References Not Discussed:**

RoSTE combines activation rotation with SFT but does not include comparisons or discussions with the following rotation-based quantization methods:

1. DuQuant: Distributing outliers via dual transformation makes stronger quantized LLMs, NeurIPS 2024.
2. OstQuant: Refining Large Language Model Quantization with Orthogonal and Scaling Transformations for Better Distribution Fitting, ICLR 2025.

Given the published date of these methods, I believe a comparison with DuQuant is necessary, and OstQuant could be a useful additional comparison.

**Experimental Designs Or Analyses:**

The paper provides a detailed comparison with state-of-the-art PTQ methods and simple Straight-Through Estimator training methods. Although the results show improvements, there remains a significant gap compared to BF16 performance.

**Methods And Evaluation Criteria:**

The evaluations include two different settings for Pythia and Llama 3.1, which effectively demonstrate the method's efficacy across different architectures.

**Other Comments Or Suggestions:**

I found the analysis and figures in the Appendix to be quite insightful, especially Figure 4, Figure 5, and Section F. It would be beneficial to incorporate some of these analyses into the main body of the paper for better accessibility.

**Other Strengths And Weaknesses:**

I observed that for Llama3.1 8B, on certain evaluation benchmarks (e.g., TruthfulQA), RoSTE performs worse than QuaRot. I recommend providing analysis and insights into the reasons behind this discrepancy.

**Questions For Authors:**

1. Could you include comparisons with the missing baselines mentioned above?
2. Could you provide more analysis on the training cost of RoSTE compared to traditional PTQ methods?
3. Why do the PTQ baselines perform poorly in Experiment 1? Is this due to the Pythia 6.9 model or the SFT dataset?
4. Please include measurements of inference speedup and memory usage.

**Relation To Broader Scientific Literature:**

The paper follows the field of rotation-based quantization methods, such as QuaRot and SpinQuant. However, it lacks a comparison with some important baselines that would further strengthen the evaluation.

**Theoretical Claims:**

I have reviewed the proof section in the Appendix, and the theoretical foundation of the method is solid.

---

> ### Author Rebuttal · Authors · 2025-04-01
>
> *Table B: Training Time and Training Memory Consumption. Our server of 8 $\times$ A100 has a total GPU memory of 320GB.*
> | Model| Method| Total Training Time (h) | Peak Memory (GB) |
> |-|-|-|-|
> | Qwen 2.5 7B  | SFT| 2.1| 300|
> || GPTQ|  2.1| 0|
> || QuaRot| 2.1 | 0|
> || SpinQuant| 3.4| 263|
> || LoRA| 0.55| 173|
> || QLoRA| 0.83| 98|
> || STE| 2.4| 317|
> || RoSTE| 2.8| 318|
>
> *Table C.1: Extended Experiment Results on Qwen 2.5 0.5B model.*
> |Bit-width| Method| ROUGE-1 | ROUGE-2 | ROUGE-L | ROUGE-LSum | ROUGE (Avg.) |
> |-|-|-|-|-|-|-|
> | **BF16**   | Base| 23.79| 6.63| 18.46| 18.56| 16.86|
> || SFT| 32.58| 11.93| 25.53| 25.55| 23.90|
> |**W4A4KV4**| QuaRot| 9.94| 0.57| 8.18| 8.38| 6.67|
> || DuQuant |  4.05 | 0.09 | 3.53 | 3.58 | 2.81 |
> || **RoSTE** | **30.75** | **10.44** | **23.96** | **23.96** | **22.28** |
> |**W4A8KV4** | QuaRot | 8.24 | 1.25 | 7.51 | 7.23 | 6.06 |
> || DuQuant | 3.91 | 0.06 | 3.56 | 3.53 | 2.77  |
> |**W4A4KV8** | QuaRot | 29.34 | 9.08 | 22.21 | 22.15 | 20.70 |
> || DuQuant | 30.22 | 10.25 | 23.17 | 23.20 | 21.71 |
>
> *Table C.2: Extended Experiment Results on Qwen 2.5 7B model.*
> |Bit-width| Method| ROUGE-1 | ROUGE-2 | ROUGE-L | ROUGE-LSum | ROUGE (Avg.) |
> |-|-|-|-|-|-|-|
> |**BF16** | Base| 32.72| 11.82 | 25.18 | 25.42 | 23.79 |
> || SFT| 34.75| 13.59 | 27.56 | 27.58 | 25.87 |
> |**W4A4KV4**| QuaRot| 7.21 | 0.10 | 5.93 | 5.93 | 4.79|
> || DuQuant |  0.00 | 0.00 | 0.00 | 0.00 | 0.00 |
> || **RoSTE** | **34.01** | **12.89** | **26.74** | **26.74** | **25.10** |
> | **W4A8KV4** | QuaRot | 5.62 | 0.15 | 5.08 | 5.14 | 3.99 |
> || DuQuant | 0.24 | 0.00 | 0.24 | 0.24 | 0.18 |
> | **W4A4KV8** | QuaRot | 31.96 | 10.98 | 24.73 | 24.88 | 23.13 |
> || DuQuant | 33.47 |12.13 |25.28 | 25.30 | 24.05 |
>
>
> > RoSTE combines activation rotation with SFT but does not include comparisons or discussions with the following rotation-based quantization methods ...
>
> We included DuQuant in the new experiments on Qwen 2.5 models in the above Table C.1, C.2. Despite our efforts spent on hyperparameter tuning (including learnable weight clipping, activation clipping, epoch, block size), we found that DuQuant suffers from a huge performance degradation in W4A4KV4 and W4A8KV4 setup.
>
> > I observed that for Llama3.1 8B, on certain evaluation benchmarks (e.g., TruthfulQA), RoSTE performs worse than QuaRot. I recommend providing analysis and insights into the reasons behind this discrepancy.
>
> We suspect that this discrepancy is due to that the two approaches use different objectives: QuaRot is not adapted to the SFT loss; RoSTE directly performs training on the SFT loss and SFT dataset. It is expected that QuaRot's performance distribution will shift further away from the full-precision SFT model. We also remark that none of the quantized model can outperform the full-prec models in the TruthfulQA benchmark.
>
> > Could you provide more analysis on the training cost of RoSTE compared to traditional PTQ methods?
>
> Details on the training cost can be found in Table B. Moreover, we illustrate in [figure](https://i.imgur.com/1kVfgSq.png) on the training cost-accuracy trade-offs of RoSTE and other SOTA methods.
>
> > Why do the PTQ baselines perform poorly in Experiment 1? Is this due to the Pythia 6.9 model or the SFT dataset?
>
> We suspect that the poor performance is due to the fine tuned Pythia models.
>
> - In the original papers, the PTQ baselines (QuaRot, SpinQuant, DuQuant) are only proposed for evaluation on pre-trained Llama models. As such, we speculate that an adaptive rotation configuration is necessary to achieve good performance for LLMs other than the Llama family.
> - Most of these methods were not evaluated on fine-tuned benchmarks. We speculate that fine tuned models may introduce certain features that are not addressed in existing PTQ methods.
>
> > Please include measurements of inference speedup and memory usage.
>
> Since our model's structure (cf. Fig 4) is equivalent to QuaRot, the inference time comparison can be referred to [Ashkboos et al., 2024b] for an extensive measurement on the inference speedup and inference memory consumption. E.g., Fig 7 therein shows that 4-bit quantized models with Hadamard rotation has 4x speedup compared to full-precision models, Fig 4 also shows significant advantages on time-to-first-token and peak memory saving. We will include these references in the revision.

---

> > ### Comment · Reviewer_mQhc · 2025-04-06
> >
> > Thank you for the detailed rebuttal. Most of my concerns have been addressed. However, I still have some confusion regarding the last question. In particular, I believe the figures from QuaRot may not be directly applicable to illustrate the inference speedup of your method. If possible, it would be helpful to provide some results based on RoSTE. Overall, I continue to recommend acceptance of this paper.

---

> > > ### Author Response · Authors · 2025-04-09
> > >
> > > Thank you for the response. We are glad that we have addressed your concerns.
> > >
> > > Furthermore, we confirm that RoSTE achieves similar inference time speedup and memory reduction performance as QuaRot, while achieving significant accuracy improvement. In Table D.1 and D.2, we evaluate the actual speedups brought by RoSTE over the *full precision fine tuned* model using the open-source setup in QuaRot's paper [https://arxiv.org/abs/2404.00456 ], modified for Llama 3.1 8B.
> > >
> > >
> > > *Table D.1: Inference Speedup and Memory Saving against Full-precision models, evaluated using W4A4KV4 RoSTE Quantized Llama 3 8B on RTX 3090 with 2048 Sequence Length on One Transformer layer in Prefilling Stage.*
> > > | Batch Size | 1 | 2 | 4 | 8 | 16
> > > |-|-|-|-|-|-|
> > > Inference Speedup (QuaRot) | 2.253x | 2.276 | 2.307x | 2.38x | 2.402x |
> > > **Inference Speedup (RoSTE)**| **2.337x** | **2.354** | **2.396x** | **2.481x** | **2.497x** |
> > > Peak Memory Saving (QuaRot) | 3.436x | 3.178x | 2.814x | 2.397x | 2.013x |
> > > **Peak Memory Saving (RoSTE)**| **3.436x** | **3.178x** | **2.814x** | **2.397x** | **2.013x** |
> > >
> > > *Table D.2: Inference Speedup and Memory Saving against Full-precision models, evaluated using W4A4KV4 RoSTE Quantized Llama 3 8B on RTX 3090 with Batch Size 1 End-to-End Decoding.*
> > > | Sequence Length | 1024 | 2048 | 4096 | 8192 |
> > > |-|-|-|-|-|
> > > Inference Speedup (QuaRot) | 1.392x | 1.614x | 1.805x | 1.831x |
> > > **Inference Speedup (RoSTE)** | **1.398x** | **1.62x** | **1.807x** | **1.839x** |
> > > Peak Memory Saving (QuaRot) | 2.874x | 2.88x | 2.892x | 2.914x |
> > > **Peak Memory Saving (RoSTE)** | **2.874x** | **2.88x** | **2.892x** | **2.914x** |
> > >
> > > We publish the modified code of QuaRot used in running the above experiment in [https://anonymous.4open.science/r/RoSTE_benchmark1 ] and will publish the trained quantized weights if the paper is accepted.
> > >
> > >
> > > The setting of the above experiments follow from **Fig. 4** of QuaRot's paper. Observe the inference speedup (2-3x) and memory usage saving (2-3x) statistics over different sequence lengths are similar to the speedup for Llama 2 7B model reported in QuaRot. This is expected since RoSTE trains a model with similar architecture to QuaRot. We also emphasize that in the meanwhile, the RoSTE trained models has much better accuracy than QuaRot model.
> > >
> > > We kindly remind the reviewer that you can update the **Overall Recommendation** score if our discussion changes your mind and you appreciate our work.

---

### Official Review · Reviewer_6QQT · 2025-03-12

**Overall Recommendation:** 3

**Summary:**

This paper aims to combine quantization-aware SFT and rotation strategy. This work is the first to leverage rotation-based quantization in QA-SFT.

The authors propose a bilevel optimization formulation – upper level subproblem for optimizing weight matrices and lower level subproblem for selecting rotation matrix.  The theoretical analysis on the benefits of rotation-enabled quantization is conducted.

The proposed method improves the performance of quantize models on downstream tasks, outperforming several baseline methods.

## update after rebuttal
The paper presents a strong theoretical analysis, especially in explaining how the proposed RoSTE method helps reduce quantization error. Meanwhile, there are notable concerns regarding its practical applicability, particularly around the resource requirements during fine-tuning. As discussed between other reviewer and the authors, the method seems to demand a substantial number of GPUs, yet the main paper lacks a clear and thorough analysis of this computational overhead.

**Claims And Evidence:**

The paper demonstrates the effectiveness of its contributions through various experimental results and in-depth theoretical analysis. However, the theoretical analysis relies on several assumptions, such as the interpolation condition and properties of the Gram matrix.

**Essential References Not Discussed:**

None

**Experimental Designs Or Analyses:**

The experiments conducted on multiple models for a single task (summarization) using both traditional fine-tuning and modern LLM fine-tuning approaches are highly commendable. Moreover, the fact that RoSTE maintains a lower quantization error compared to STE is a notable advantage of the method.

**Methods And Evaluation Criteria:**

The paper dedicates significant effort to justify the superiority of its method. It presents extensive evaluations, showing fine-tuning performance using RoSTE on two models across a variety of tasks. In particular, for LLaMA, the experiments cover a more general fine-tuning case which covers a wide range of and tasks, while for Pythia, multiple metrics are reported when fine-tuning on a summarization dataset. This results in a wealth of experimental evidence supporting the method's effectiveness.

**Other Comments Or Suggestions:**

Fine-tuning on the Tulu-3 SFT-mixture dataset closely resembles the fine-tuning approaches commonly used with modern LLMs. It would be beneficial to see experimental results on models other than LLaMA or on smaller models, as this could further validate the generalizability and efficiency of the method across different model scales and architectures.

**Other Strengths And Weaknesses:**

**Strength**: Superior performance compared to baselines

**Weakness**: Performance variation on models (Pythia and LLaMA)

**Questions For Authors:**

- Can you provide fine-tuning results of smaller models using Tulu-3 SFT dataset?

- It seems that there are far more cases where a model that has already been fine-tuned needs to be quantized. Can the QA-SFT technique be practically applied in various situations?

**Relation To Broader Scientific Literature:**

The paper builds on and is influenced by prior work on rotation-based post-training quantization (PTQ) methods and research on the straight-through estimator (STE). In particular, it takes cues from earlier studies that used rotation techniques to mitigate the impact of outlier activations and improve quantization accuracy, and from foundational works on STE that addressed gradient approximation issues during quantization-aware training.

**Theoretical Claims:**

The proofs generally follow established techniques and seem to be correct. However, I did not rigorously verify every detail, and it is worth noting that the theoretical claims depend on certain assumptions (such as the interpolation condition and specific properties of the Gram matrix) that could limit their applicability in some scenarios.

---

> ### Author Rebuttal · Authors · 2025-04-01
>
> We sincerely appreciate the reviewer for acknowledging the strength of our approach. Below we summarize our response to your concerns.
>
> > it is worth noting that the theoretical claims depend on certain assumptions (such as the interpolation condition and specific properties of the Gram matrix) that could limit their applicability in some scenarios.
>
> We agree. However, we point out that the analysis in Sec 4 aims only at giving *theoretical insights* for the design of RoSTE algo, rather than ensuring convergence for STE training on LLMs, which is an open problem as modern LLM involves complex architecture. Despite the added assumptions, we retain essential elements of RoSTE training with linear layer featuring a rotation matrix in activations and weights, and the derived results gives motivation for RoSTE to use quantization error in the lower level objective of (11). To our best knowledge, this is the first analysis that provides insights on the convergence of STE training with rotation. Our derived bound echoes with the observation in Table 2 and Figure 3, where RoSTE empirically outperforms STE without rotation on quantization error and downstream task accuracy; also see the new experiments in Table A.1, A.2, A.3, A.4 in the response to Rev. c8mS.
>
> > Can you provide fine-tuning results of smaller models using Tulu-3 SFT dataset?
>
> Since Tulu 3 is originally proposed for Llama 3.1 base models, extending the application of Tulu-3 SFT dataset to other families of base models would require an extensive fine-tuning on the hyperparameters. Due to limited time, we cannot produce additional results on Tulu-3 dataset. We will include additional results in the revision.
>
> > It seems that there are far more cases where a model that has already been fine-tuned needs to be quantized. Can the QA-SFT technique be practically applied in various situations?
>
> It is correct that there are many unquantized but fine-tuned model that are publicly available. The proposed RoSTE can be applied on them through either (i) knowledge distillation, e.g., LLM-QAT in [Liu et al., 2023], or (ii) treating them as initialization using the original fine-tuning dataset. Note that all our numerical results indicate that existing PTQ methods may fail to maintain the fine-tuned accuracy after quantization. Our work thus demonstrate to practitioners that a good QA-SFT strategy (e.g., RoSTE) can greatly improve the quality of quantized & fine-tuned model.

---

### Official Review · Reviewer_c8mS · 2025-03-15

**Overall Recommendation:** 3

**Summary:**

RoSTE introduces a novel Quantization-Aware Supervised Fine-Tuning (QA-SFT) approach for large language models (LLMs), addressing the inefficiencies of traditional post-training quantization (PTQ) and the high computational cost of quantization-aware training (QAT). The proposed Rotated Straight-Through Estimator (RoSTE) integrates rotation-based quantization into QA-SFT, leveraging an adaptive rotation strategy using Walsh-Hadamard matrices to mitigate outliers in activations and weights. Theoretical analysis demonstrates that RoSTE effectively reduces quantization error, leading to improved performance over existing PTQ and QAT methods. Experimental results on Pythia and Llama models confirm that RoSTE achieves superior accuracy in 4-bit quantized models while maintaining computational efficiency.

**Claims And Evidence:**

The claim that *"Applying the theorem shows that given \( R \), the resultant prediction error of the intermediate model \( w_T \) will be bounded by \( O\left(\sum_{s=0}^{T}(1-\mu)^{T - s}\mathbb{E}[\|Q_w(Rw_s) - Rw_s\|_G^2]\right) \),"* seems problematic for a few reasons.

The theorem is derived under a highly simplified setting that assumes a quadratic loss and a linear model. However, modern Transformer-based large language models (LLMs) have complex architectures involving non-linearity, residual connections, and multi-head attention mechanisms. The direct application of this theorem to such models is questionable, as it does not account for the intricate dependencies and interactions within deep networks. Moreover, the proof relies on the interpolation assumption (Assumption 4.2), which states that for any rotation matrix \( R \), there exists an interpolating weight \( w^*_R \) such that the model can perfectly predict the target. In reality, this assumption is unlikely to hold in large-scale LLMs trained on diverse datasets. At last, the proposed rotation matrix optimization method, which relies on randomized Walsh-Hadamard transformations, lacks rigorous justification regarding its effectiveness in minimizing quantization error in a structured and optimal way.

The claim regarding the prediction error bound is not convincingly supported by theoretical or experimental evidence. The analysis is based on unrealistic assumptions, and the experimental validation does not directly confirm the theoretical findings. Therefore, the authors should either provide stronger empirical evidence to justify their claim or acknowledge the limitations of their theoretical analysis in real-world LLM scenarios.

**Essential References Not Discussed:**

N/A

**Experimental Designs Or Analyses:**

The paper does not provide sufficient analysis of the computational overhead introduced by RoSTE. The method involves additional operations such as rotation transformations using Walsh-Hadamard matrices, which may introduce extra latency during training and inference. However, the paper does not quantify the trade-offs between accuracy improvements and the increased computational cost. Given that efficiency is a primary motivation for quantization, failing to report potential slowdowns or extra memory requirements makes it difficult to assess whether RoSTE is truly practical for real-world deployment.

While RoSTE is positioned as an efficient quantization-aware fine-tuning approach, the paper does not provide detailed benchmarks on key efficiency metrics, such as GPU memory consumption, inference latency, or speedup comparisons against standard PTQ/QAT baselines. Since quantization is primarily used to reduce computational and memory overhead, these factors should be explicitly reported. Without these insights, it is unclear whether RoSTE is actually more practical than existing quantization methods when deployed on hardware-constrained environments.

**Methods And Evaluation Criteria:**

The paper primarily evaluates RoSTE on Pythia (1B/6.9B) and Llama (8B) models, which, while relevant, do not provide sufficient diversity in terms of model architectures and scales. Given that LLM quantization techniques should generalize across different model families (e.g., GPT, Mistral, Falcon, or transformer variants with different architectural choices), the narrow selection raises concerns about the method’s broader applicability.

**Other Comments Or Suggestions:**

N/A.

**Other Strengths And Weaknesses:**

The paper presents an original and insightful approach to quantization-aware fine-tuning by integrating rotation-based quantization with theoretical justification. The idea of linking quantization error and fine-tuned model error is a notable contribution, as it provides a more structured understanding of quantization effects in LLMs. Additionally, the use of Hadamard and identity matrix combinations for adaptive rotation selection is a novel and efficient way to address activation outliers, making the approach both practical and theoretically grounded.

One notable weakness is the lack of clarity and organization in the writing. The paper feels somewhat dense and unpolished, with certain sections being overly wordy and difficult to follow. This can make it challenging for readers to extract key insights efficiently. In particular, some theoretical derivations and algorithmic descriptions could be streamlined for better readability. Furthermore, while the work presents strong theoretical contributions, the practical aspects of computational overhead and real-world deployment feasibility remain underexplored.

**Questions For Authors:**

Please refer to the aforementioned comments.

**Relation To Broader Scientific Literature:**

The key contributions of this paper are well-grounded in the broader scientific literature on QAT and rotation-based quantization for LLMs. One of the most significant contributions of this work is its theoretical analysis, which establishes a direct connection between quantization error (particularly from weight and activation quantization) and the final prediction error of fine-tuned models. While previous works have studied the impact of quantization on model accuracy (e.g., GPTQ [Frantar et al., 2022], LLM-QAT [Liu et al., 2023]), this paper goes further by providing a formal mathematical framework to quantify this relationship. This insight contributes to a deeper understanding of how fine-tuned models behave under aggressive quantization, particularly in the 4-bit regime.

Another intriguing aspect of the paper is its approach to constructing rotation matrices using a combination of Hadamard and identity matrices. Rotation-based quantization methods have been explored in prior work (e.g., QuaRot [Ashkboos et al., 2024], SpinQuant [Liu et al., 2024]), but most of these methods focus on static rotation applied during PTQ. In contrast, this paper integrates adaptive rotation selection into the QA-SFT process, which is a novel extension. The idea of using Hadamard transformations to selectively apply rotation layer-by-layer is particularly interesting because it balances the benefits of reducing outliers while maintaining computational efficiency.

**Theoretical Claims:**

I have already addressed the issue of the claims relying on overly strong assumptions.

---

> ### Author Rebuttal · Authors · 2025-04-01
>
> *Table A.1: Results on Pythia 1B model.*
> | Bit-width  | Method| ROUGE-1 | ROUGE-2 | ROUGE-L | ROUGE-LSum | ROUGE (Avg.) |
> |-|-|-|-|-|-|-|
> | **FP16**| Base| 22.40| 5.73| 17.35| 17.59| 15.77|
> || SFT| 32.80| 11.84| 25.49| 25.50| 23.91|
> | **W4A4KV4**| **RoSTE** | **31.80** | **11.03** | **24.71** | **24.71** | **23.07** |
> |$r = 64$ | QLoRA| 22.58 | 5.87 | 17.48 | 17.71 | 15.91|
>
> *Table A.2: Results on Pythia 6.9B model.*
> | Bit-width| Method| ROUGE-1 | ROUGE-2 | ROUGE-L | ROUGE-LSum | ROUGE (Avg.) |
> |-|-|-|-|-|-|-|
> |**FP16**| Base| 28.81 | 9.45 | 22.29 | 22.91 | 20.87 |
> || SFT| 33.69 | 12.60 | 26.27 | 26.31 | 24.72 |
> |**W4A4KV4**| **RoSTE** | **32.60** | **11.54** | **25.22** | **25.25** | **23.66** |
> |$r = 64$ | QLoRA  | 27.92 | 8.91 | 21.97 | 22.00 | 20.20 |
>
> *Table A.3: Results on Qwen 2.5 0.5B model.*
> |Bit-width|Method| ROUGE-1 | ROUGE-2 | ROUGE-L | ROUGE-LSum | ROUGE (Avg.) |
> |-|-|-|-|-|-|-|
> | **BF16**| Base| 23.79| 6.63| 18.46| 18.56| 16.86|
> || SFT| 32.58| 11.93   | 25.53| 25.55| 23.90|
> | **W4A4KV4**| RTN| 10.04   | 0.37    | 8.15| 8.34| 6.73|
> || GPTQ| 12.53| 0.92    | 10.08| 10.50| 8.51|
> || QuaRot| 9.94| 0.57| 8.18| 8.38| 6.67|
> || SpinQuant| 12.16| 1.22| 10.69| 10.72| 8.70|
> |$r = 64$| QLoRA | 24.88 | 7.18 | 19.28 | 19.43 | 17.69 |
> || STE| 29.97| 9.92| 23.39   | 23.39| 21.67|
> || **RoSTE** | **30.75** | **10.44** | **23.96** | **23.96** | **22.28** |
>
> *Table A.4: Results on Qwen 2.5 7B model.*
> | Bit-width | Method | ROUGE-1 | ROUGE-2 | ROUGE-L | ROUGE-LSum | ROUGE (Avg.) |
> |-|-|-|-|-|-|-|
> | **BF16** |Base| 32.72|11.82| 25.18 |25.42|23.79|
> || SFT| 34.75|13.59|27.56|27.58| 25.87 |
> | **W4A4KV4** |RTN|1.07|0.00| 1.01|1.01|0.77|
> ||GPTQ| 0.72|0.00|0.69|0.69| 0.53|
> ||QuaRot|7.21|0.10|5.93|5.93| 4.79|
> ||SpinQuant|6.87|0.29|5.97|6.12|4.81|
> |$r = 64$|QLoRA|32.22|11.41|24.75|24.89|23.32|
> ||STE|30.86|10.16|23.73|23.73|22.12|
> ||**RoSTE**|**34.01** | **12.89** | **26.74** | **26.74** | **25.10** |
>
> > The authors should either provide stronger empirical evidence to justify their claim or acknowledge the limitations of their theoretical analysis in real-world LLM scenarios.
>
> We agree with your points. However, we point out that the analysis in Sec 4 aims at giving *theoretical insights* for the design of RoSTE algo, rather than ensuring convergence for STE training on LLMs, which is an open problem to our best knowledge as modern LLM involves complex architecture. Despite the limitations raised, we retain essential elements of RoSTE with a linear layer featuring a rotation matrix in activations and weights, and the derived results motivated RoSTE to minimize quantization error in the lower level objective (11). The derived bound echoes with our Table 2 and Fig 3. We will emphasize these limitations in the revision.
>
> > primarily evaluates RoSTE on Pythia (1B/6.9B) and Llama (8B) ...
>
> We extended our experiments to the latest Qwen 2.5 models - see Table A.3, A.4 on accuracy of 4-bit quantized 0.5B / 7B models. RoSTE delivers the top quantized model vs the baselines.
>
> > Given that efficiency is a primary motivation...  quantify the trade-offs between accuracy improvements and the increased computational cost.
>
> We agree, but would clarify that there are two aspects of efficiency wrt computational cost - inference and training, which we will detail below:
>
> - *Inference efficiency*: Since our model's structure (cf. Fig 4) is equivalent to QuaRot which implements weight and activation quantization with Hadamard rotation, the inference time comparison can be referred to [Ashkboos et al., 2024b] for an extensive measurement on the inference speedup and inference memory consumption. E.g., Fig 7 therein shows that 4-bit quantized models with Hadamard rotation has 4x speedup compared to full-precision models, their Fig 4 also shows significant advantages on time-to-first-token and peak memory saving.
> - *Training efficiency*: As we restricted the search space of rotation matrices to Hadamard matrix, the theoretical training complexity should be on par with full-param SFT. This is confirmed in the additional Table B of the response to Reviewer mQhc. We acknowledge that the training complexity can still be higher than parameter-efficient methods such as QLoRA. For a comprehensive comparison, we refer to [figure](https://i.imgur.com/1kVfgSq.png) on the accuracy-vs-training-time tradeoff for SOTA methods. Observe that RoSTE achieves the best accuracy at a moderate cost in training complexity.
>
> We will include the above references in the revision.
>
> > the paper does not provide detailed benchmarks on key efficiency metrics ...
>
> We apologize for the misunderstanding caused by ambiguous wording of the title. We have referred to "efficiency" in the context of deployment for trained quantized model, where RoSTE delivers models with low latency inference and memory consumption. Nevertheless, in the training phase, RoSTE is on par with full-param SFT as seen in Table B in the response to Rev mQhc.

---

> > ### Comment · Reviewer_c8mS · 2025-04-08
> >
> > The rebuttal has partially addressed my concerns and questions; however, I remain somewhat skeptical about the claimed performance and memory benefits of the proposed scheme. Therefore, I will keep my original score.

---

> > > ### Author Response · Authors · 2025-04-09
> > >
> > > Thank you for the response. We are glad that some of your concerns have been addressed.
> > >
> > > To address your skepticism on the efficiency improvement by RoSTE, we confirm that RoSTE achieves similar inference time speedup and memory reduction performance as QuaRot, while achieving significant accuracy improvement. In Table D.1 and D.2, we evaluate the actual speedups brought by RoSTE over the *full precision fine tuned* model using the open-source setup in QuaRot's paper [https://arxiv.org/abs/2404.00456 ], modified for Llama 3.1 8B.
> > >
> > > *Table D.1: Inference Speedup and Memory Saving against Full-precision model, evaluated using W4A4KV4 RoSTE Quantized Llama 3 8B on RTX 3090 with 2048 Sequence Length on One Transformer layer in Prefilling Stage.*
> > > | Batch Size | 1 | 2 | 4 | 8 | 16
> > > |-|-|-|-|-|-|
> > > Inference Speedup (QuaRot) | 2.253x | 2.276 | 2.307x | 2.38x | 2.402x |
> > > **Inference Speedup (RoSTE)**| **2.337x** | **2.354** | **2.396x** | **2.481x** | **2.497x** |
> > > Peak Memory Saving (QuaRot) | 3.436x | 3.178x | 2.814x | 2.397x | 2.013x |
> > > **Peak Memory Saving (RoSTE)**| **3.436x** | **3.178x** | **2.814x** | **2.397x** | **2.013x** |
> > >
> > > *Table D.2: Inference Speedup and Memory Saving against Full-precision model, evaluated using W4A4KV4 RoSTE Quantized Llama 3 8B on RTX 3090 with Batch Size 1 End-to-End Decoding.*
> > > | Sequence Length | 1024 | 2048 | 4096 | 8192 |
> > > |-|-|-|-|-|
> > > Inference Speedup (QuaRot) | 1.392x | 1.614x | 1.805x | 1.831x |
> > > **Inference Speedup (RoSTE)** | **1.398x** | **1.62x** | **1.807x** | **1.839x** |
> > > Peak Memory Saving (QuaRot) | 2.874x | 2.88x | 2.892x | 2.914x |
> > > **Peak Memory Saving (RoSTE)** | **2.874x** | **2.88x** | **2.892x** | **2.914x** |
> > >
> > >
> > > We publish the modified code of QuaRot used in running the above experiment in [https://anonymous.4open.science/r/RoSTE_benchmark1 ] and will publish the trained quantized weights as well if the paper is accepted.
> > >
> > > Note that detailed setup of the above experiment follows from **Fig. 4** of QuaRot's paper. Observe that RoSTE achieves similar inference speedup (2-3x) and memory saving (2-3x) to those reported in QuaRot's paper for Llama 2 7B model. This is expected since RoSTE trains a model with similar architecture to QuaRot. We also emphasize that in the meanwhile, the RoSTE trained models has much better accuracy than QuaRot model.
> > >
> > > With the new experiments demonstrated above and together with the previous comment on the training complexity compared to full param SFT (see Table B in response to **Rev. mQhc**), we believe that there is sufficient evidence demonstrating the efficiency of RoSTE in both training and inference.
> > >
> > > We emphasize that our main contribution is to derive a novel method for fine tuning LLMs on an **inference efficient architecture with incoherence processing**, i.e., the QuaRot's architecture. In fact, from our experiments, we found that **many PTQ methods will fail outside of the Llama family**, regardless of the inference efficient architecture used, on a broad range of fine-tuning benchmarks. This motivated our paper to propose a **model agnostic** QAT method, i.e., performing QAT using adaptive rotation to achieve better downstream task performance through a theoretically supported training procedure over different models and data. Through joint training and adaptive rotation, we can maximally take into consideration data and model properties. We believe that these contributions are both novel and significant in a practical sense.
> > >
> > > We kindly remind the reviewer that you can update the **Overall Recommendation** score if our discussion changes your mind and you appreciate our work.

---

### Official Review · Reviewer_W2sL · 2025-03-23

**Overall Recommendation:** 2

**Summary:**

This paper introduces RoSTE, a method for quantization-aware SFT of LLMs. RoSTE aims to jointly optimize model weights and rotation matrices during fine-tuning, enabling efficient 4-bit quantization of weights, activations, and KV caches in a single training phase. This integration contrasts with the approaches that apply quantization after fine-tuning models, which can degrade performance. The core idea of RoSTE is to leverage a bi-level optimization approach: (1) updating model weights using a rotation-aware STE, and (2) selecting rotation matrices from a candidate set to minimize a quantization error surrogate loss. The authors show the effectiveness of their proposal by providing theoretical analysis and empirical evidences.

**Claims And Evidence:**

While the idea of jointly optimizing rotation matrices and quantized weights is compelling and likely beneficial for the quantized SFT model quality, I see several limitations in the current approach.

1. Memory Efficiency: The proposed method relies on QAT, which is memory-intensive and difficult to scale to large models (e.g., 70B+). This contrasts with qLoRA-based approaches that are significantly more memory-efficient by freezing most of the model and only training low-rank adapters. RoSTE adds further overhead by optimizing rotation matrices, increasing the computational overhead.
2. Limited Learning Space: The rotation search is restricted to identity and Hadamard matrices, which is a highly constrained design choice. Prior work (e.g., SpinQuant) has shown that Hadamard rotations are not optimal, and more flexible, learned rotations can yield better results. From the limited ablation study, it seems like learning the rotation matrix does not contribute much towards the model quality.
3. Empirical Evaluation: The paper claims state-of-the-art performance, but the empirical evidence does not fully support this. Most comparisons are against post-training quantization (PTQ) methods, while stronger baselines such as qLoRA or other parameter-efficient quantization-aware fine-tuning approaches are omitted.

**Essential References Not Discussed:**

N/A

**Experimental Designs Or Analyses:**

Some potential issues with the experiments:
1. Missing analysis of memory usage and training time.
2. Scalability to larger models (70B+) is not demonstrated or discussed.
3. No comparison with qLoRA or other PEFT baselines.
4. Some experiments use outdated, undertrained models; newer, more competitive models should be considered more.

**Methods And Evaluation Criteria:**

The motivation to jointly optimize quantized parameters and rotation matrices is well-founded. However, the practicality of the proposed approach remains unclear--in particular, whether it can scale to larger models. Some experiments are conducted on relatively old models (Pythia) that are considered undertrained by today’s standards, and the baselines are mostly limited to post-training quantization methods, omitting more competitive approaches such as qLoRA-based fine-tuning.

**Other Comments Or Suggestions:**

N/A

**Other Strengths And Weaknesses:**

Some weaknesses of the paper are:
1. Novelty is limited--the paper combines existing ideas (QAT, STE, rotation-based quantization), but does not introduce fundamentally new techniques.
2. Scalability and efficiency are not thoroughly demonstrated, especially for large models.
3. Empirical results are not fully convincing — baseline comparisons are limited, and stronger methods like qLoRA are missing.
4. While the integration is well-motivated, the overall significance and impact remain unclear without broader evaluation.

**Questions For Authors:**

Please see other sections.

**Relation To Broader Scientific Literature:**

RoSTE builds upon previous approaches (rotation-based PTQ, QAT, and STE) and proposes to combine them to achieve better quantized SFT fine-tuning performance. While the idea of combining QAT with rotation is new in this context, the method largely builds on known techniques like STE, Hadamard rotations, and existing QAT frameworks, extending them to the supervised fine-tuning setting. However, the paper does not compare against more recent memory-efficient fine-tuning methods like qLoRA, which have gained traction in the literature.

**Theoretical Claims:**

The theoretical claims seem correct.

---

> ### Author Rebuttal · Authors · 2025-04-01
>
> > Memory Efficiency: ...
>
> We agree partially with your points. While RoSTE maybe less memory efficient than methods such as QLoRA, we observe that RoSTE has a similar training cost in compute and memory usage as full-param SFT (cf. Table B in response to Rev. mQhc). The overhead with adaptive rotation is insignificant due to the small outer iteration no.
>
> Our goal lies in achieving optimal performance at *deployment* by tuning fully quantized models with inference speedup and downstream task accuracy. As a comparison, note that Table 10 of [arxiv/2404.14047] shows models such as QLoRA suffer from degraded inference speed due to extra unquantized LoRA layers.
>
> From a practical standpoint, our experiments on <8B models is relevant to on-device deployment where LLMs of such scales are common. E.g., the fine tuned 4-bit Pythia 1B model by RoSTE has greatly improved performance over PTQ on the full-precision base model. This will be useful for applications that prioritize inference latency.
>
> For 70B+ models, RoSTE is still feasible and can potentially improve model quality. First, a rough estimate suggests that using RoSTE to training Llama3 70B under Adam require ~900GB of GPU memory (see [here](https://community.ibm.com/community/user/cloud/blogs/arindam-dasgupta/2024/09/18/calculating-gpu-requirements-for-efficient-llama-3)), which is implementable on a 8xH200 cluster. Second, recent studies such as [arxiv/2407.11062] showed that using QAT on 70B+ models with 1xA100 is possible with a simple block processing technique. Such technique can be straightforwardly combined with RoSTE.
>
> > Limited Learning Space: ...
>
> This is a good point. Searching for optimal rotation like SpinQuant may enhance the model quality, yet our design to limit the learning space to $\\{I, H\\}$ for each layer is grounded on the tradeoff of complexity vs performance:
>
> - *Quantization Error*: As shown in Fig 3, using Hadamard matrices suffices to reduce the quantization error and optimize the lower-level objective in (11);
> - *Training*: extending the search space to all rotation matrices leads (11) to become a mix of non-smooth and manifold optimization problems. Handling it requires substantially higher training complexity.
> - *Inference*: Hadamard matrices are efficient for inference [Ashkboos et al., 2024b], where it supports hardware acceleration and has low memory footprint due to its integer structure. In contrast, optimal rotation requires real-valued rotation matrices that impose significant overheads during inference.
>
> Our design choice is supported by results in Table 2 & 3: RoSTE outperforms learnt rotations (SpinQuant) & no-rotation (STE).
>
> > Empirical Evaluation: ...
>
> We extended our experiment results to QLoRA in the above Table A.1, A.2, A.3, A.4 in the response to Rev. c8mS. Notice that QLoRA under-performs RoSTE in terms of quantized model accuracy on downstream tasks, potentially due to the limited learning space of low-rank adaptation.
>
> > Missing analysis of memory usage and training time.
>
> We included details on training time and training memory usage in Table B in the response to Rev. mQhc. Notice that RoSTE has a similar training cost as STE.
>
> > Scalability to larger models (70B+) is not demonstrated or discussed.
>
> As mentioned in the first response, scaling RoSTE to 70B+ models is feasible through either
> - Scaling up computation capability.
> - Applying block processing to RoSTE.
>
> While we do not have the computation resource nor time to experiment with 70B+ models at the moment, our [codebase](https://anonymous.4open.science/r/RoSTE) is available for the community to run further experiments.
>
> > Some experiments use outdated ...
> > Broader eval needed ...
>
> We extended our experiment results to Qwen 2.5 models (released on Sep 25 2024). By Table A in the above, we illustrate the accuracy of 4-bit quantized Qwen 2.5 7B models. RoSTE remains to produce the top performance quantized model among the baselines. Now, our evaluation set covers three families of LLMs (Pythia, Qwen, Llama). Our coverage is broader than existing literature such as QuaRot and SpinQuant, where only Llama models were evaluated.
>
> > Novelty is limited ...
>
> We respectfully disagree. From a methodology perspective, our work is the first to employ adaptive rotation with SFT to improve the accuracy of quantized models. Such design is well grounded in theory from an optimization perspective.
>
> Besides innovation in methodology, we observe a research gap where prior works mainly considered pre-trained zero-shot evaluation, while the performance of quantized fine-tuned models remains unexplored. Our results open the door to a new benchmark for approaches that directly optimize the SFT objective on quantized models. We further illustrate the existing research gap in [figure](https://i.imgur.com/1kVfgSq.png) on accuracy-vs-training-time tradeoffs. Note that RoSTE achieves better accuracy than existing methods on SFT benchmark with a moderate training overhead.

---

### Decision · Program_Chairs · 2025-05-01

**Decision:**

Accept (poster)

**Comment:**

This paper proposes a new quantization-aware supervised fine-tuning method with efficiency by leveraging adaptive rotation strategy to reduce activation outliers for large langage models.

This paper has mixed scores such as three acceptance and one weak reject.

The main concerns were speedup effects, memory usages, and large requirement of GPU resources.

During the rebuttal, some major concerns seemd to be addressed, and GPU requirement issue still remains.

For clear decision, AC discussed with the reviewers. AC also agree that 2 nodes of H100 is quite high requirement compared to PEFT appraoch such as QLoRA. However, it is natural that PEFT's GPU requirement is much smaller than that of SFT. For SFT, 2 nodes are typical and practical cases in real-world LLM SFT applications.

Therefore, AC recommends accpeting this paper and asks the authors to clarify this issue.